



# The water column of the Yamal tundra lakes as a microbial filter preventing methane emission

Alexander Savvichev[1], Igor Rusanov[1], Yury Dvornikov[2,3], Vitaly Kadnikov[1], Anna Kallistova[1], Elena Veslopolova[1], Antonina Chetverova[4,5], Marina Leibman[3,6], Pavel Sigalevich[1], Nikolay Pimenov[1], Nikolai Ravin[1], and Artem Khomutov[3,6]

[1]Winogradsky Institute of Microbiology and Institute of Bioengineering, Research Centre of Biotechnology of the Russian Academy of Sciences, Moscow, 119071, Russia
[2]Peoples' Friendship University of Russia, Moscow, 117198, Russia
[3]Earth Cryosphere Institute of Tyumen Scientific Centre, Siberian Branch, Russian Academy of Sciences, Tyumen, 625000, Russia
[4]Institute of Earth Sciences, Saint-Petersburg State University, Saint-Petersburg, 199034, Russia
[5]Otto-Schmidt Laboratory of Arctic and Antarctic Research Institute, Saint-Petersburg, 199397, Russia
[6]University of Tyumen, International Institute of Cryology and Cryosophy, Tyumen, 625003, Russia

*Correspondence to*: Pavel A. Sigalevich (pavelsigalevich@list.ru)

**Abstract.** Microbiological, molecular ecological, biogeochemical, and isotope geochemical research was carried out in four lakes of the central part of the Yamal Peninsula in the area of continuous permafrost. Two of them were large (73.6 and 118.6 ha) and deep (up to 10.6 and 12.3 m) mature lakes embedded into all geomorphological levels of the peninsula, and two others were smaller (3.2 and 4.2 ha) shallow (up to 2.3 and 1.8 m) lakes which appeared as a result of thermokarst on constitutional (segregated) ground ice. Samples were collected in August 2019. The Yamal tundra lakes were found to exhibit high phytoplankton production ($340-1200$ mg C m$^{-2}$ day$^{-1}$) during the short summer season. Allochthonous and autochthonous, particulate and dissolved organic matter was deposited to the bottom sediments, where methane production occurred due to anaerobic degradation ($90-1000$ µmol CH$_4$ dm$^{-3}$). The rates of hydrogenotrophic methanogenesis appeared to be higher in the sediments of deep lakes than in those of the shallow ones. In the sediments of all lakes, *Methanoregula* and *Methanosaeta* were predominant components of the archaeal methanogenic community. Methane oxidation ($1.4-9.9$ µmol dm$^{-3}$ day$^{-1}$) occurred in the upper sediment layers simultaneously with methanogenesis. *Methylobacter tundripaludum* (family *Methylococcaceae*) predominated in the methanotrophic community of the sediments and the water column. The activity of methanotrophic bacteria in deep mature lakes resulted in a decrease of the dissolved methane concentration in lake water from $0.8-4.1$ µmol CH$_4$ L$^{-1}$ to 0.4 µmol CH$_4$ L$^{-1}$, while in shallow thermokarst lakes the geochemical effect of methanotrophs was much less pronounced. Thus, only small shallow Yamal lakes may contribute significantly to the overall diffusive methane emissions from the water surface during the warm summer season. The water column of large deep lakes on Yamal acts, however, as a microbial filter preventing methane emission into the atmosphere.



# 1 Introduction

Climate warming, recorded on the Earth in recent decades, is especially pronounced at high latitudes of the Northern Hemisphere (IPCC, 2014). As a result, ground temperatures of the permafrost-covered area undergo a consistent rise (Biskaborn et al., 2019). The Yamal peninsula is a remarkable area characterized by active cryogenic relief-forming processes (Kizyakov and Leibman, 2016) and many water bodies. Limnicity of Yamal varies between 10 and 20% depending on the position at the geomorphological level with maximum observed on floodplains (Romanenko, 1999). Classification of most of those lakes as thermokarst lakes of continuous ice-rich permafrost has been suggested (Dubikov, 1982) although other origins have also been proposed (Arctic and Antarctic Research Institute, 1977; Kritsuk, 2010). Thermokarst lakes originate from the thawing of ice-rich permafrost or pure ice of various genesis (Kachurin, 1961). This thawing process results in topographic depressions that are immediately filled with water in case if the topography of the area is flat (Romanovskii, 1993). Thermokarst lakes are widespread in West and East Siberia, in Alaska and Northern Scandinavia (Grosse et al., 2013; Kravtsova and Rodionova, 2016; Vonk et al., 2015; Wik et al., 2016). The depth of these lakes is highly dependent on the type of ground ice beneath. Shallow lakes may result from the thaw of segregated ice with lower ice content (Dostovalov and Kudryavtsev, 1967) than i.e. tabular ground ice widespread in the North of West Siberia. These lakes are generally shallow (< 3.0 m) and range in size from a few square meters to hundreds of square kilometers (Grosse et al., 2013; Wik et al., 2016).

For the area of continuous permafrost in West Siberia, the increase of total lake area by 12% is observed in recent decades due to climate warming (Smith et al., 2005). Although small thermokarst lakes were found to be more active greenhouse gas emitters than large lakes, their contribution to the total diffusive greenhouse gas emissions for the entire West Siberian Lowland is considered to be minor, less than 1–1.5% (Polishchuk et al., 2018). Nonetheless, the rates of diffuse methane emission from some thermokarst lakes are rather high, exceeding the emission from terrestrial tundra ecosystems (Wik et al., 2016). The amount of methane emitted to the atmosphere is expected to be increasing due to the permafrost thaw (Heslop et al., 2015; Laurion et al., 2010; Martinez-Cruz et al., 2015; Pokrovskiy et al., 2012; Serikova et al., 2019; Townsend-Small et al., 2017; Vonk et al., 2015; Walter Anthony et al., 2007; Wik et al., 2016). Permafrost thawing results in an inflow of biogenic elements as components of mineral and organic matter (OM) into freshwater lakes (thermokarst, floodplain, gas-emission craters (GEC), etc.) (Dvornikov et al., 2018; Vonk et al., 2015). In summer mineral compounds stimulate phytoplankton development (blooms) in the photic layer of the water column (Edelstein et al., 2017). Cyanobacterial bloom results in the formation of autochthonous organic matter (primary production) (Patova, 2014). The newly formed organic matter acts as an easily consumable trophic resource for heterotrophic bacterioplankton. Autochthonous and allochthonous organic matter forms a suspension in water, which subsequently precipitates forming the bottom sediments. OM degradation in the sediments occurs as a result of activity of a psychrophilic microbial community. Since sulfate concentration in Yamal lakes is rather low (Fotiev, 1999), sulfate reduction is not intense, and methane produced by methanogenic archaea is the main product of OM mineralization (Sepulveda-Jauregui et al., 2015; Vonk et al., 2015; Walter Anthony et al., 2006). Data on methane emission from different types of lakes, as well as on the ratio between modern microbially produced methane and the methane from thawing permafrost,



are important for the understanding of the methane cycle in this area. Methane is actively consumed by methanotrophic microorganisms, under both oxic and anoxic conditions. Methanotrophs, as well as methanogens, play a key role in the methane cycle; they oxidize this greenhouse gas and decrease methane emission into the atmosphere. Methane, whether it is released from thawing permafrost deposits or produced in the sediments by methanogenic archaea, arrives into the water column. In the lakes located in the permafrost area, methanotrophs consume up to 60% of total methane (Singleton et al., 2018; Xu et al.,

2016). The highest rates of methane oxidation are usually revealed in the top layer of the sediments, where $CH_4$ and $O_2$ form steep counter gradients (Auman et al., 2000). Hence, by mitigating $CH_4$ emissions, microorganisms act as an efficient microbial $CH_4$ filter (Bastviken et al., 2004; Cole et al., 1994).

The carbon isotope composition of methane ($\delta^{13}$C) varies depending on its origin (Sassen and Macdonald, 1997). In the course of oxidation by methanotrophic bacteria, methane containing the light carbon isotope is preferentially consumed (Colin Murrell

and Jetten, 2009; Hamdan et al., 2011). Methane carbon enriched with the light $^{12}$C isotope is converted to $CO_2$ and organic compounds in the cells of methanotrophs. Thus, OM of methanotrophic origin enriches suspended organic matter with the light carbon isotope, while methane unconsumed by methanotrophs becomes enriched with the heavy carbon isotope. The data on methane carbon isotope composition indicate therefore the geochemical consequences of microbial processes of the methane cycle (Heuer et al., 2009).

In Yamal, large deep lakes are widely presented, especially on floodplains of rivers (such as Mordy-Yakha and Se-Yakha). Their basins are highly developed, and it can be considered that the thermokarst is presently not the main process involved in the formation of such lake basins. One more mechanism responsible for the emergence of new lakes in Yamal is the formation of gas-emission craters. Recent studies related to these permafrost features (Leibman et al., 2014) showed that initially deep (20–50 m) craters rapidly (within three to four summer seasons) turned into shallow (3–5 m) lakes by filling the craters with

thawed tabular ground ice and atmospheric precipitations (Dvornikov et al., 2019). These newly formed lakes become very similar to other lakes in Yamal in terms of their morphometry and hydrochemistry. Winter studies revealed lower methane production in the sediments of young lakes filling the gas-emission craters than in the sediments of mature tundra lakes (Savvichev et al., 2018a).

The present work was aimed at elucidation of the similarities and differences in the rates of the methane cycle processes and

the composition of the relevant microbial communities between small young lakes (constitutional ice thermokarst) and deep mature lakes (massive ground ice thermokarst).

## 2 Sampling site and analytical methods

### 2.1 Study site

In this work, four basins were under study: two typical shallow thermokarst lakes and two large and deep lakes located in the

vicinity of Vas'kiny Dachi research station in Central Yamal (Fig. 1), within the framework of the long-term monitoring program of the Earth Cryosphere Institute (Dvornikov et al., 2016). All studied lakes are located in the area of continuous



permafrost with the average ground temperature of up to –7°C at the level of zero annual amplitudes and with the active layer depth varying between 0.5 and 1.3 m. The permafrost is characterized by high ice content (Dubikov, 1982). The morphometric characteristics of the studied lakes are summarized in Table 1.

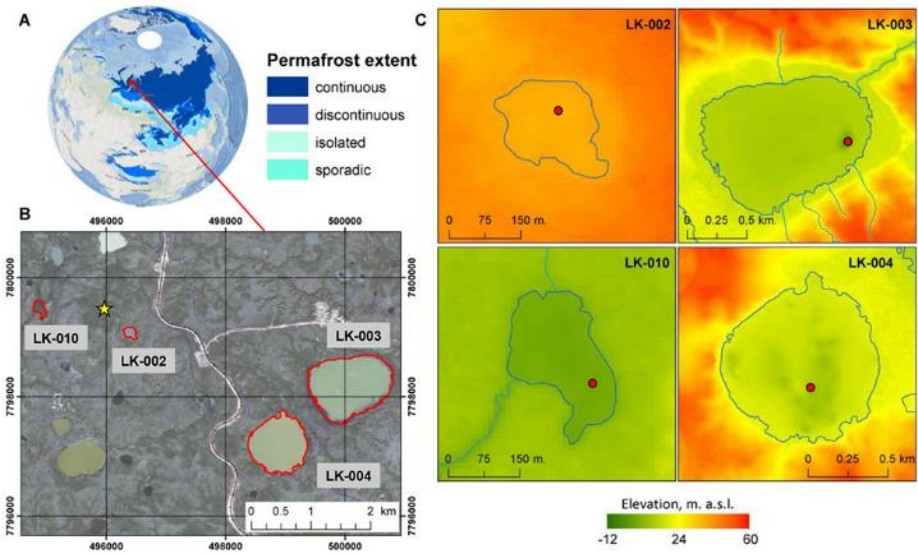


**Fig. 1. A: An overview map of Eurasia, permafrost extent from (Brown et al., 2002). B: Study area map of Central Yamal, with red outlined polygons indicating the lakes under study, yellow star – Vas'kiny Dachi research station, the orthorectified QuickBird satellite image acquired July 30, 2010 as a background (source: Digital Globe Foundation ©), datum – WGS-84, projection – UTM**
**Zone 42N. C: the topography of the lake basins under study, red dots indicate locations of the water and sediment samples, elevations are given in the Baltic height system (1977).**

**Table 1. Characteristics of Yamal lakes, morphometry, hydrological and hydrochemical parameters.**

| Lake_ID, (depth, m) | Type, aging of lakes | Basin embedded into | Area, ha | NL WL | EC, µS sm$^{-1}$ | O$_2$, mg L$^{-1}$ | T° C | Secci depth, m | Sampling horizons, m |
|---|---|---|---|---|---|---|---|---|---|
| | | | | | Surface water (Near-bottom water) | | | | |
| LK-010 (1.2) | T, modern | Floodplain | 4.25 | 70.3012 68.8642 | 269 | 9.4 | 14.4 | 0.90 | 0; 1.1 |
| LK-002 (2.0) | T, modern | IV Coastal-marine plain | 3.23 | 70.2977 68.9045 | 118 | 9.8 | 11.7 | 0.75 | 0; 0.9; 1.9 |
| LK-003 (10.6) | NT, mature | V Marine plain | 118.6 | 70.2898 69.0019 | 123 (124) | 10.0 (8.6) | 15.8 (12.0) | 1.20 | 0; 2.5; 7.0; 10.0 |





| LK-004 (11.5 | NT, mature | V Marine plain | 73.6 | 70.2809 68.9705 | 115 (116) | 9.7 (8.6) | 15.1 (14.2) | 1.00 | 0; 3.0; 11.0 |


The basins in the study area were mostly freshwater lakes. The predominant anion was $Cl^-$ (56.7 eq % on average among ~30 lakes), with the proportion of anions in the row: $Cl^- > HCO_3^- > SO_4^{2-}$. Cations were strongly dominated by $Na^+ + K^+$ (58.5 eq% on average). The proportion of cations was in the sequence: $Na^+ + K^+ > Mg^{2+} > Ca^{2+}$. The total mineralization in lakes was normally lower than 150 mg $L^{-1}$ (Dvornikov et al., 2019).

The lakes LK-002 and LK-010 are small in area, with the depth not exceeding 2.3 m (Table 1). During winter, most of the area of these lakes is frozen to the bottom. Their bottom sediments have a high content of undecomposed organic matter. The lakes LK-003 and LK-004 were large, deep and had well-developed basins with sandy bottoms. According to their basic morphometric characteristics, we suggested lakes LK-002 and LK-010 to be the young, developing lakes of thermokarst origin, while lakes LK-003 and LK-004 being mature, of debatable origin.

**2.2 Sample Collection and Characterization**

Water and surface sediment samples were collected from four lakes during the period from August 5 till August 9, 2019. Water samples were collected from the entire water column at the deepest parts of the lake (Table 1). A total of 13 water samples were collected: LK-003 − surface, 2.5, 7.0, and 10.0 m (near the bottom); LK-004 − surface, 3.0, and 11.0 m (near the bottom); LK-002 − surface, 0.9, and 1.9 m (near the bottom); LK-010 − surface and 1.1 m (near the bottom). Water was sampled using

a TD-Automatika© hydrological water sampler, dispensed into 35-mL glass vials, sealed with gas-tight rubber stoppers (avoiding gas bubbles), and covered with a perforated aluminum cap. Bottom sediments (with a core length of up to 340 mm) were collected using a limnologic stratometer with a glass tube. A total of 12 sediment samples were collected: LK-003, 0−3, 3−7, and 3−12 cm; LK-004, 0−4, 4−9 and 9−15 cm; LK-002, 0−4, 4−9, and 9−14 cm; LK-010, 0−3, 3−6, and 6−12 cm. The thawed ground ice flowing into the lake (C) was collected at the lower part of the lake LK-004 slope.

Sediment samples were then transferred into cut-off 5-mL plastic syringes, preserving the structure of the sediment core, and sealed with gas-tight rubber stoppers. Samples of water and sediments were stored in a portable temperature-controlled box at +8 to +12°C. Pore water from sediment samples was obtained by centrifugation at 8000 *g* for 10 min in a TsUM-1 centrifuge (Russia).

The temperature, electrical conductivity, and concentration of dissolved oxygen were measured using a WTW 3320 SET2

portable multimeter (Germany) equipped with a CellOx-325 dissolved oxygen detector and a TetraCon 325 conductometer.

**2.3 Analytical Techniques**

Methane content in the water and sediment samples was determined using the head-space method (McAuliffe, 1971). Methane concentration was measured on a Kristall-2000-M gas chromatograph (Chromatec, Yoshkar-Ola, Russia) equipped with a





flame ionization detector. The concentration of sulfate and chloride ions was determined (after distillation and concentration)

on a Staier ion chromatograph (Akvilon, Russia). Carbon content in dissolved organic matter (DOC) was determined by high-temperature incineration (non-purgeable organic carbon, NPOC) using a Shimadzu TOC-Vcph analyzer (Japan), with the measurement accuracy of ±10%. Average values were calculated from three samples. Statistical treatment of the results was performed using Excel 2000.

### 2.4 Bacterial Abundance

For assessment of total microbial abundance (MA) and microbial biomass, water samples in glass vials were fixed with glutaraldehyde at the final concentration of 2%. Fixed samples (5–10 mL) were filtered through black polycarbonate 0.2-μm filters (Millipore, USA). The filters were stained with acridine orange (2 mg mL$^{-1}$) (Hobbie et al., 1977) and examined under an Olympus BX 41 epifluorescence microscope equipped with an Image Scope Color (M) visualization system. Cell volumes of the cocci and rods were calculated by approximating them as geometric spheres and cylinders, respectively. The cells were

enumerated in 20 fields of view.

### 2.5    Radiotracer Experiments

The rates of microbial processes: light and dark $CO_2$ assimilation (LCA and DCA), autotrophic (hydrogenotrophic) methanogenesis (MG), and methane oxidation (MO) were determined using radiotracer labeled compounds: $NaH^{14}CO_3$, specific activity 2.04 GBq mmol$^{-1}$, Amersham, UK (10 μCi per sample) and $^{14}CH_4$, specific activity 1.16 GBq mmol$^{-1}$, JSC

Isotope, Russia (1 μCi per sample). To determine the LCA and DCA rates, two transparent and one darkened vials were used for each sampling horizon. Each transparent vial was covered with an individual sheath calibrated for transmission of the photosynthetically active radiation corresponding to illumination in the sampling horizon. A labeled substrate (0.2 mL as a sterile degassed water solution) was injected through the rubber stopper with a syringe. The vials were incubated for half the daylight period at in situ temperature. After incubation, they were fixed with 1 mL 0.1 N HCl and filtered through 0.2-μm

nylon membranes. Photosynthetic production was calculated as the difference between the values for the dark and transparent vials. Incubation of water and sediment samples to determine the rates of other processes was also carried out in situ. The microbial processes (MG, MO) were stopped by injecting 0.5 mL of saturated KOH solution into each experimental vial. All experiments were performed in duplicate. After the end of the experiments, the vials were stored at 5–10°C. Measurement of the radioactivity of the products of microbial activity in both the experimental and control vials was performed in the laboratory

according to methods described earlier (Pimenov and Bonch-Osmolovskaya, 2006; Savvichev et al., 2018b). Radioactivity was measured on a TRI-Carb TR 2400 scintillation counter (Packard, USA). Calculation of the LCA and DCA rates were done considering the $^{14}C$-$CO_2$ both in bacterial cells and in the extracellular dissolved organic matter. For calculation of MO rates, $^{14}C$-$CH_4$ conversion to $CO_2$, biomass, and extracellular soluble OM were analyzed separately. The confidence interval for the LCA, DCA, MG, and MO rates varied from 10 to 40%.



### 2.6 Isotopic Composition of Methane Carbon

The δ¹³C methane value was determined on a Delta Plus mass spectrometer (Thermo Electron Corporation, Langenselbold, Germany), using a PDB-calibrated standard, and calculated using the following equation:

$$\delta^{13}C = ([^{13}C] / [^{12}C])sample / ([^{13}C] / [^{12}C])standard - 1) \times 1000‰ \tag{1},$$

where $([^{13}C] / [^{12}C])$sample / $([^{13}C] / [^{12}C])$standard are the ratios of occurrence of the $^{12}C$ and $^{13}C$ atoms in the sample and in the standard, respectively. The international PDB standard used has the isotope occurrence ratio $[^{13}C] / [^{12}C]$ of 0.001172 (Craig, 1957). For methane, $\delta^{13}C\text{-}CH_4$ was measured on a TRACE GC gas chromatograph (Thermo Fisher Scientific, USA) coupled to a Delta Plus mass spectrometer. The error of the δ¹³C measurements did not exceed ±0.1‰.

### 2.7 DNA Extraction and Sequencing Procedure

To collect microbial biomass, the water sample (500 mL) was passed through filters with a pore diameter of 0.22 μm. The filters were homogenized by triturating with liquid nitrogen, and the preparation of metagenomic DNA was isolated using the DNeasy PowerSoil Kit (Qiagen, Hilden, Germany) according to the manufacturer's instructions. The total amount of isolated DNA was about 0.5 μg for each sample. The V3–V4 variable region of the prokaryotic 16S rRNA genes was obtained by PCR with the primers 341F (5'- CCTAYGGGDBGCWSCAG) and 806R (5'- GGACTACNVGGGTHTCTAAT) (Frey et al., 2016). PCR fragments were barcoded using Nextera XT Index Kit v2 (Illumina, USA). The PCR fragments were purified using Agencourt AMPure Beads (Beckman Coulter, Brea, CA, USA) and quantitated using Qubit dsDNA HS Assay Kit (Invitrogen, Carlsbad, CA, USA). Then all the amplicons were pooled together in equimolar amounts and sequenced on the Illumina MiSeq instrument (2 × 300 nt reads). Paired overlapping reads were merged using FLASH (Magoč and Salzberg, 2011).

### 2.8 Bio-informatic Analysis

The pool of 16S rRNA gene sequences was analyzed with QIIME 2 v.2019.10 (https://qiime2.org) (Caporaso et al., 2010). DADA2 plugin was used for sequence quality control, denoising and chimera filtering (Callahan et al., 2016) Operational Taxonomic Units (OTUs) were clustered applying VSEARCH plugin (Rognes et al., 2016) with open-reference function using Silva v. 132 database (Glöckner et al., 2017; Quast et al., 2013) with 97% identity. Taxonomy assignment was performed using BLAST against Silva v. 132 database with 97% identity.

### 2.9 Nucleotide Sequence Accession Number

The raw data generated from 16S rRNA gene sequencing were deposited in Sequence Read Archive (SRA) under the accession numbers SRR11972844 -SRP266728, available via BioProject PRJNA636944.





# 3 Results

## 3.1 Net Primary Production (PP), Dissolved Organic Carbon (DOC), Microbial Abundance (MA) and Dark Carbon Assimilation (DCA) in the water column

Phytoplankton primary production (PP) is the main biogeochemical (ecological) characteristic of water bodies. According to our data, PP in all studied lakes was relatively high (Fig. 1).

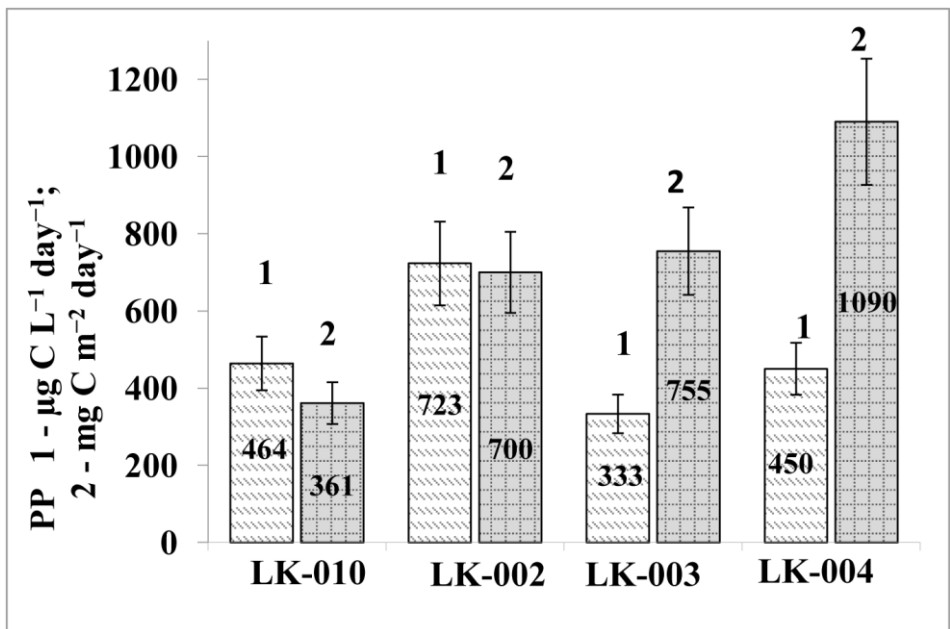

**Fig. 2. Primary production in four lakes of the Yamal Peninsula (August 2019): in the upper water horizon, µg C L⁻¹ day⁻¹ (1), and integral PP for the photic layer, mg C m⁻² day⁻¹ (2).**


While PP values for the surface layers of the lakes LK-003, LK-004, and LK-010 were similar, PP in lake LK-002 was considerably higher (Fig. 2). PP calculation for the photic depth (i.e., for the 1 m² water column; mg C m⁻² day⁻¹) revealed, however, higher PP in two deep mature lakes (Fig. 2).

DOC concentration in the water column did not vary significantly within each lake ($\pm 0.5$ mg L⁻¹). Large deep lakes were
characterized by slightly lower DOC (4.5–5.5 mg L⁻¹) than small shallow lakes (7–8 mg L⁻¹). Upper horizons of lake sediments had much higher DOC (10–40 mg L⁻¹) and higher variability within the sediment column (Fig. 3).



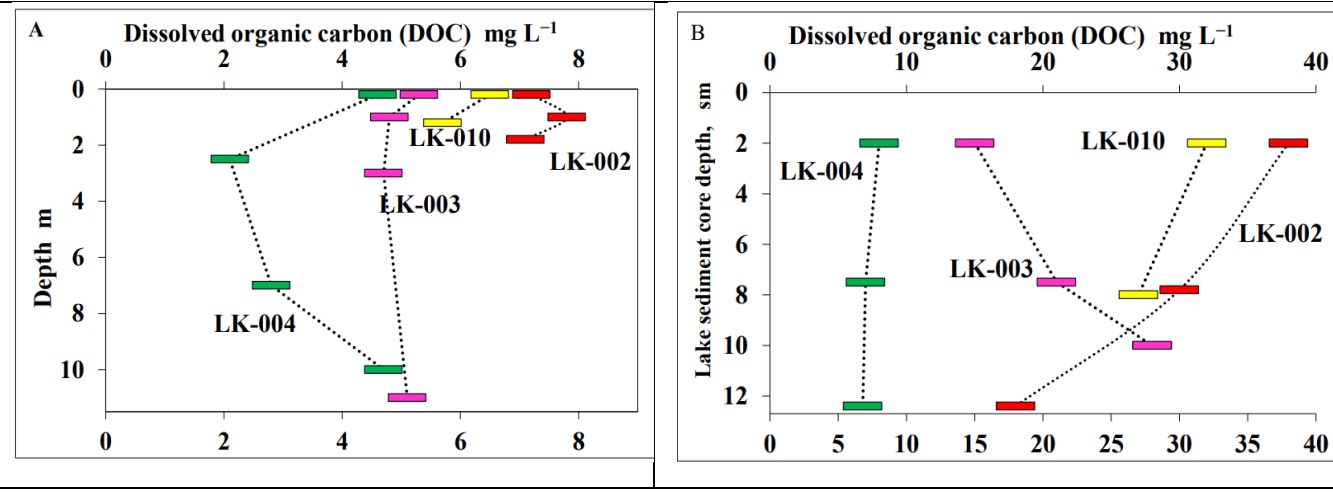

**Fig. 3. DOC concentration in the water column (a) and bottom sediments (b) of four lakes.**


Microscopic analysis of the water column samples revealed relatively high microbial abundance ($1.7-7.6 \times 10^6$ cells mL$^{-1}$, typical of mesotrophic water bodies (Table 2).

**Table 2. Microbial abundance (MA), cell volume, and biomass (B) in the water column of the studied Yamal lakes**

| Lake | Depth, m | MA, $10^6$ cells mL$^{-1}$ | Cell volume (average), µm$^3$ | B, µg L$^{-1}$ | Aggregated cells, % |
|---|---|---|---|---|---|
| LK-010 | Surface  (w1) | $1.7 \pm 0.2$ | 0.12 | $220 \pm 20$ | 1.8 |
|  | 1.0–1.1  (w2) | $2.3 \pm 0.2$ | 0.13 | $300 \pm 20$ | 2.0 |
| LK-002 | Surface  (w1) | $4.6 \pm 0.4$ | 0.15 | $690 \pm 60$ | 11 |
|  | 0.9    (w2) | $3.7 \pm 0.3$ | 0.13 | $480 \pm 40$ | 10 |
|  | 1.8–1.9   (w3) | $7.6 \pm 0.4$ | 0.15 | $930 \pm 60$ | 13 |
| LK-003 | Surface  (w1) | $3.2 \pm 0.2$ | 0.15 | $520 \pm 40$ | 10 |
|  | 2.5   (w2) | $3.3 \pm 0.2$ | 0.14 | $460 \pm 40$ | 8 |
|  | 7.0   (w3) | $3.8 \pm 0.3$ | 0.12 | $450 \pm 30$ | 6 |
|  | 10.0   (w4) | $4.6 \pm 0.4$ | 0.13 | $690 \pm 50$ | 11 |





| LK-004 | Surface (w1) | 2.9 ± 0.2 | 0.13 | 390 ± 30 | 2.1 |
|---|---|---|---|---|---|
| | 3.0   (w2) | 3.5 ± 0.3 | 0.10 | 400 ± 30 | 2.0 |
| | 11.0   (w3) | 4.2 ± 0.4 | 0.11 | 490 ± 40 | 2.0 |


Despite the strong wind conditions, which caused the mixing of the water masses, MA varied slightly depending on the sampling horizon. In general, it was higher in the surface and near-bottom horizons. Both the highest MA and biomass were revealed in the near-bottom horizon of the shallow thermokarst lake LK-002, while the lowest occurred in the surface horizon of the shallow thermokarst lake LK-010. The average cell volume in the studied samples varied from 0.10 to 0.15 $\mu m^3$, and

the biomass was 220 to 930 $\mu g\ L^{-1}$. The share of aggregated cells varied from 1.8% to 11%. According to our results on total MA, all studied lakes were mesotrophic. However this assumption is based on the data obtained during the highest seasonal warmth of the water column.

Dark $CO_2$ assimilation (DCA) is an integral characteristic of the activity of chemoautotrophic and chemoheterotrophic microorganisms. The DCA values obtained in the present work varied from 23.6 $\mu g\ C\ L^{-1}\ day^{-1}$ in the surface layer of the

deep lake LK-004 to 10.8 $\mu g\ C\ L^{-1}\ day^{-1}$ in the near-bottom layer of the thermokarst lake (Fig. 4).

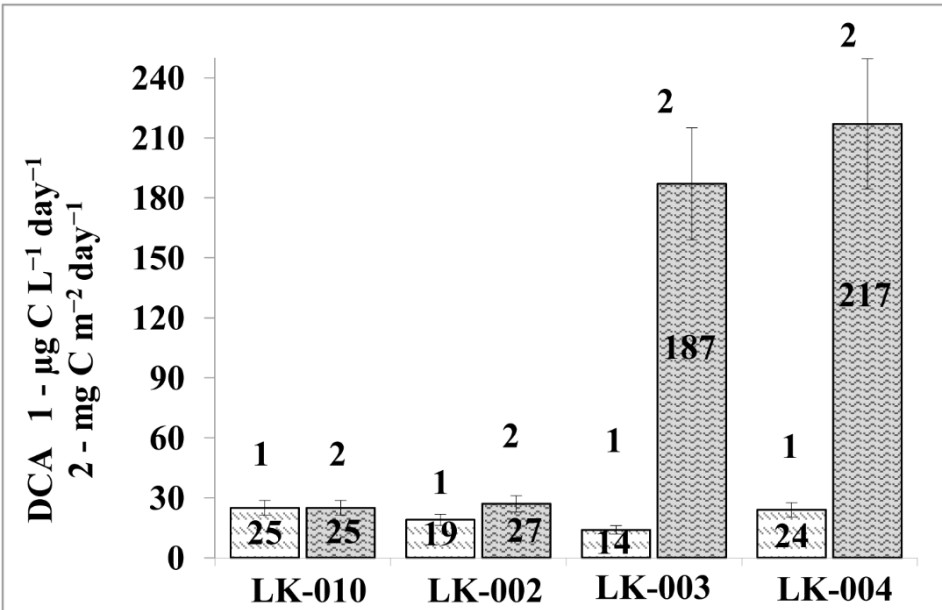

**Fig. 4  Dark $CO_2$ assimilation (DCA) in four lakes of the Yamal Peninsula (August 2019): in the upper water horizon, $\mu g\ C\ L^{-1}\ day^{-1}$ (1), and integral $DCA_{int}$ for the water column, $mg\ C\ m^{-2}\ day^{-1}$ (2).**



DCA values in the surface layers of all four lakes were quite similar (14.4−25 μg C $L^{-1}$ $day^{-1}$), which follows from similar hydrological and production characteristics. However, the $DCA_{int}$, calculated for the entire depth of the lake, shows that the values of $DCA_{int}$ in deep mature lakes are significantly higher than in shallow thermokarst lakes (187−217 mg C $m^{-2}$ $day^{-1}$ compared to 24−26 mg C $m^{-2}$ $day^{-1}$). High $DCA_{int}$ indices in deep mature lakes are determined by active heterotrophic processes that take place in all layers of the water column. It follows from the above that the depth of the lakes is an important

factor determining the degree of transformation of the organic matter in the particulate substance forming bottom sediments.

### 3.2  Dissolved Methane Content [$CH_4$] and Methane Oxidation Rate (MO)

The concentration of dissolved methane measured in the water columns of shallow, well-mixed thermokarst lakes was relatively low (310–480 nmol $CH_4$ $L^{-1}$), with no significant differences between the surface and near-bottom horizons (Fig. 5).

In deep mature lakes, methane concentrations decreased significantly (two- to tenfold) from the near-bottom horizons (4140–975 nmol $CH_4$ $L^{-1}$) up to the surface layers (400–420 nmol $CH_4$ $L^{-1}$) (Fig. 5).

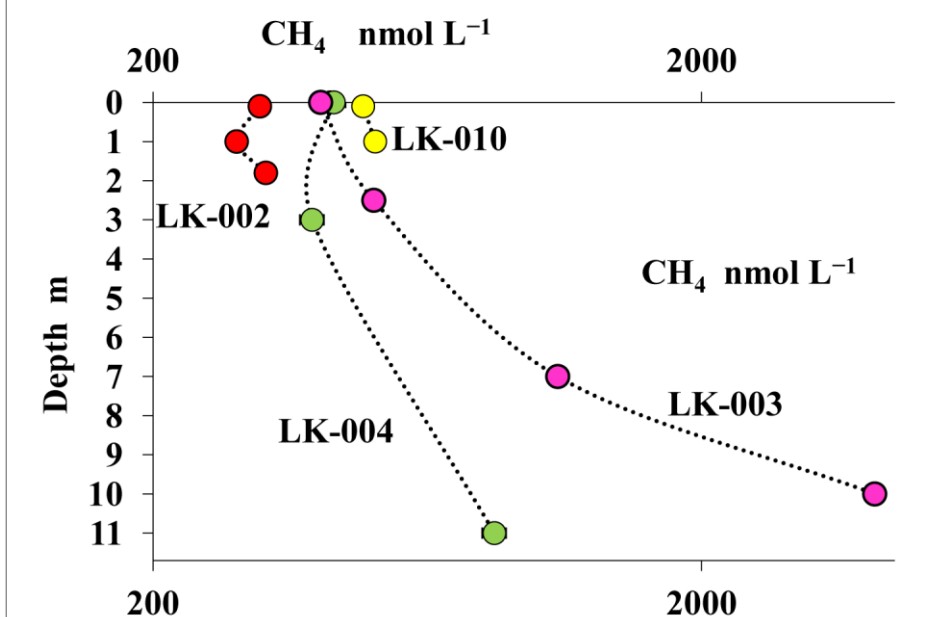

**Fig. 5. Concentrations of dissolved methane ($CH_4$, nmol $L^{-1}$) in the water columns of four Yamal tundra lakes: LK-002 (red), LK-010 (yellow), LK-003 (purple), and LK-004 (green).**

Experiments with labeled methane revealed that MO rates in the shallow, well-mixed thermokarst lakes were almost the same in the surface layer and near the bottom, although the absolute values of MO rates in the surface layers of shallow lakes LK-002 and LK-010 differed significantly (3.6 and 16.4 nmol $CH_4$ $L^{-1}$ $day^{-1}$, respectively) (Fig. 6). In deeper lakes MO rates peaked in the near-bottom horizons: 14.7 and 55.3 nmol $CH_4$ $L^{-1}$ $day^{-1}$ for the lakes LK-004 and LK-003, respectively. MO





rates decreased significantly in the upper layers of the water column correlating well with decreasing concentrations of

dissolved methane (Fig. 5).

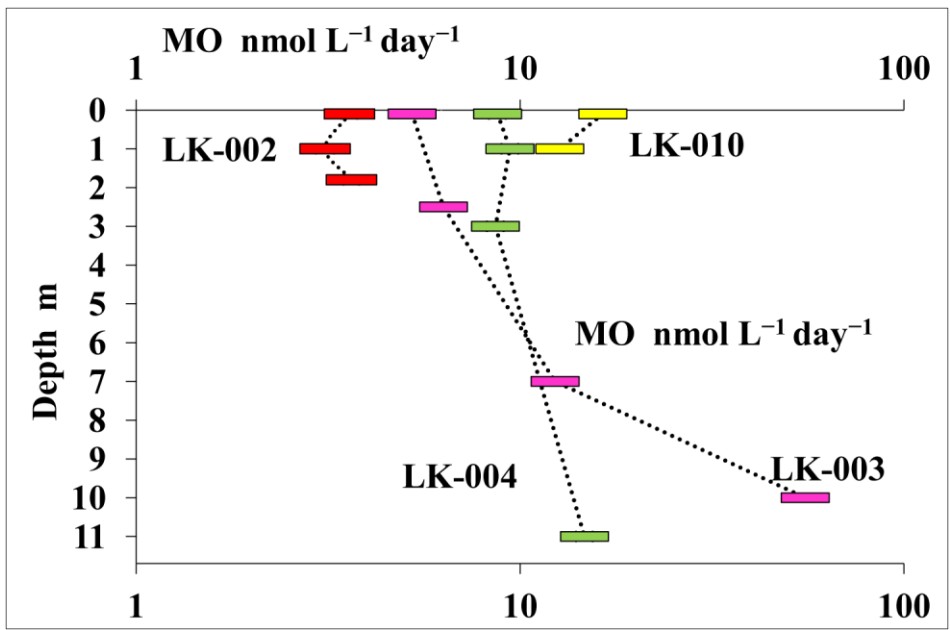

Fig. 6. Rates of methane oxidation, nmol CH$_4$ L$^{-1}$ day$^{-1}$, in the water column of four Yamal tundra lakes: LK-002 (red), LK-010 (yellow), LK-003 (purple), and LK-004 (green).

**3.3 Microbial Processes of the Methane Cycle and Carbon Isotopic Composition of Methane in the Bottom Sediments**

Bottom sediments were collected in the bottom surface and to the depth of 14−15 cm (Table 3).

**Table 3. Methane content, rates of methanogenesis (MG) and methane oxidation (MO), and methane carbon isotope composition (δ$^{13}$C-CH$_4$ ‰) in the sediments of four Yamal tundra lakes**

| Lake | Sediment depth, cm | CH$_4$, μmol dm$^{-3}$ | MG, nmol dm$^{-3}$ day$^{-1}$ | MO, μmol dm$^{-3}$ day$^{-1}$ | δ$^{13}$C-CH$_4$, ‰ |
|---|---|---|---|---|---|
| LK-010 | 0–3  (s1) | 32.8 | 20 ± 2.0 | 1.7 ± 0.2 | -69.0 |
|  | 3–6  (s2) | 291 | 27 ± 3.0 | 5.5 ± 0.6 | -69.6 |
|  | 6–12  (s3) | 365 | 10 ± 1.0 | 5.6 ± 0.6 | -68.2 |
| LK-002 | 0–4  (s1) | 93.7 | 5.3 ± 1.1 | 1.4 ± 0.2 | -80.8 |
|  | 4–9  (s2) | 235 | 4.2 ± 1.0 | 1.7 ± 0.2 | -74.8 |
|  | 9–14  (s3) | 376 | 3.7 ± 1.0 | 2.8 ± 0.3 | -77.1 |
| LK-003 | 0–3  (s1) | 594 | 35 ± 4 | 4.9 ± 0.5 | -64.9 |





| | 3–7 (s2) | 978 | 67 ± 7 | 9.8 ± 1.0 | -63.3 |
|---|---|---|---|---|---|
| | 7–12 (s3) | 986 | 203 ± 20 | 9.9 ± 1.0 | -63.5 |
| LK-004 | 0–4 (s1) | 10.8 | 9.7 ± 1.0 | 0.3 ± 0.03 | ND |
| | 4–9 (s2) | 132 | 29 ± 3.0 | 1.3 ± 0.1 | -73.4 |
| | 9–15 (s3) | 407 | 19 ± 2.0 | 2.5 ± 0.3 | -72.2 |
| LK-004 K* | 1–6 | 9.7 | 2.5± 0.4 | 0.1± 0.01 | ND |

K*, thawed ground ice from the coastal slope of lake LK-004.

ND, methane content below the analytical threshold.

The sediments of all studied lakes were similar in structure, with a loose, reddish upper horizon (to 3-4 cm depth), a denser intermediate layer of gray with some reddish interlayers, and a gray, dense lower layer. Numerous gas bubbles were visible in the sediments of lake LK-003. All samples, including the LK-003-C thawed ground ice, had pelito-aleuritic texture. Methane

content in the sediments varied from 10 to 1000 μmol $CH_4$ $dm^{-3}$ (Table 3). Rate of hydrogenotrophic (autotrophic) methanogenesis in the upper sediment layer varied within a broad range: from 3–7 to 203 nmol $dm^{-3}$ $day^{-1}$. The highest MG rate was observed in the lowermost collected sample from the deep lake LK-003, while the lowest one occurred in the lowermost horizon from the shallow thermokarst lake LK-002. The rate of methane oxidation (up to 9.9 μmol $dm^{-3}$ $day^{-1}$) in the sediments exceeded the MG rates significantly. MO was more active in the dense subsurface layer than in the loose upper

sediment layer. Methane carbon isotope composition ($\delta^{13}$C-$CH_4$) in the sediments varied from -80.8 to -63.3‰. The highest content of the light carbon isotope ($^{12}$C) in methane was observed in the lake LK-002. In the sediments of LK-003, methane contained much more heavy carbon ($^{13}$C). According to the average values of $\delta^{13}$C-$CH_4$, the studied sediments form the following sequence: LK-002 – LK-004 – LK-010 – LK-003.

### 3.4 Taxonomic Composition of Microbial Communities in the Water Column and Bottom Sediments

Analysis of the 16S rRNA gene sequences from 13 water samples and 13 sediment samples revealed high taxonomic diversity of microbial communities in all four studied lakes (Fig. 7).



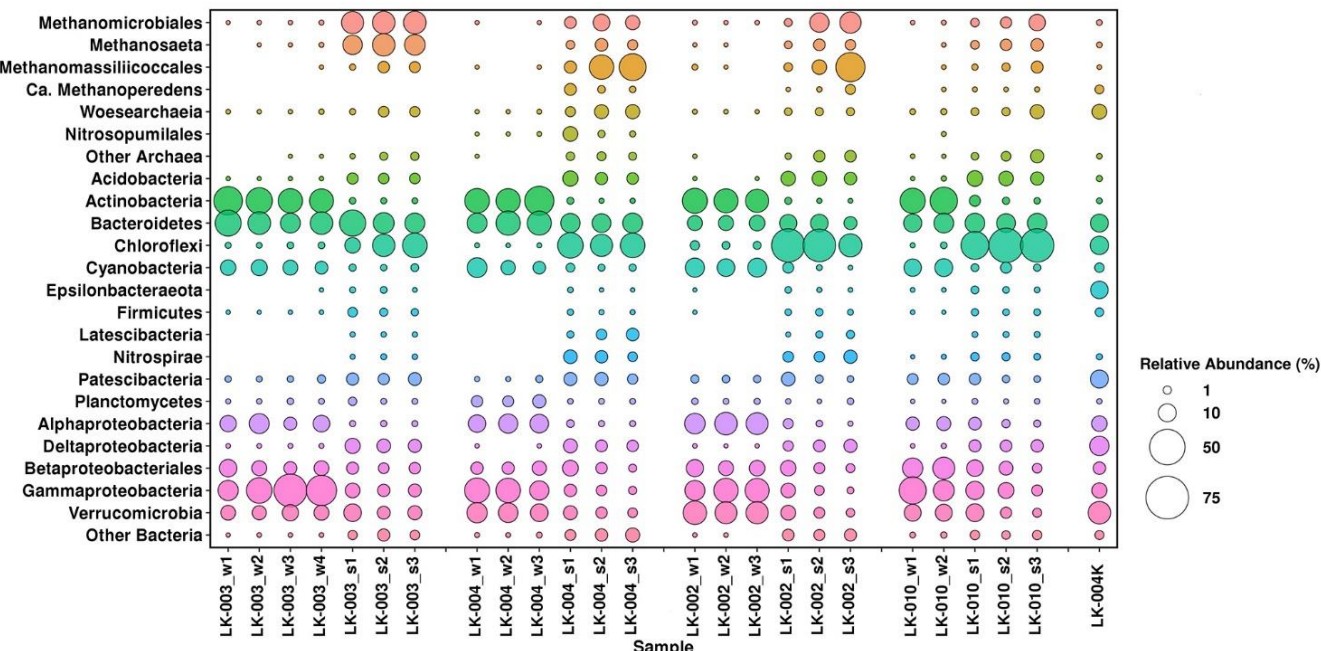

**Fig. 7. Microbial communities of the water column and bottom sediments in four tundra Lakes of Yamal peninsula determined by**
**high-throughput sequencing of 16S rRNA genes.**

The water column microbial communities of all four lakes were rather similar. Most microorganisms belonged to the Bacteria domain (99.42–99.95% of the total read number). Among them, Gammaproteobacteria (11.4–41.6%), Actinobacteria (18.4–30.7%), Bacteroidetes (6.2–24.4%), Verrucomicrobia (5.4–19.1%), Alphaproteobacteria (4.1–17.3%), and Betaproteobacteria (3.2–16.5%) were the most abundant groups. The share of cyanobacterial OTUs in the surface horizons was 6.7 to 12.4%.
Other bacterial taxa (phyla Acidobacteria, Chloroflexi, Firmicutes, Latescibacteria, Nitrospirae, Patescibacteria, Epsilonproteobacteria, and Deltaproteobacteria) constituted a minor part of the water column microbial communities in all four lakes, with the share of each taxon below 0.5%.

Microbial communities of the bottom sediments differed significantly from those of the water column. Archaea were relatively abundant in the sediments of all four lakes (20.2 – 55.6% of the total number of reads) and were mainly represented by
methanogens of the orders Methanomicrobiales, Methanosarcinales, and Methanomassiliicoccales. The upper sediment of lake LK-002 was exceptional in this respect, with the share of archaea not exceeding 3.96%, probably due to contamination of the sample with near-bottom water. This suggestion is supported by the elevated content of cyanobacteria in this sample. Bacteria of the bottom sediments of the studied lakes belonged to the phyla Chloroflexi (18.5–43.8% in thermokarst lakes and 6.6–23.2% in non-thermokarst ones), Bacteroidetes (4.2–12.6%), Verrucomicrobia (1.8–9.0%), Acidobacteria (2.3–6.5%),
Gammaproteobacteria (0.5–8.4%), Deltaproteobacteria (2.1–6.1%), Betaproteobacteria (3.2–7.2% in the upper and 0.9–3.2% in the lower horizons of all lakes), and Nitrospirae (1.7–4.8% in the lakes LK-002 and LK-004. Other bacteria revealed in the





sediments, belonging to Actinobacteria, Epsilonbacteraeota, Firmicutes, Latescibacteria, Planctomycetes, and Alphaproteobacteria, constituted less than 1.0% of the microbial community.

The composition of the community from the thawed permafrost sample (LK-003 K) differed significantly from the
communities of both the water column and the sediments. Only in this sample the "Ca. Woesearchaeota" OTUs (5.8%) were more numerous than Euryarchaeota (1.7%). Members of the phylum Epsilonbacteraeota (genera *Arcobacter* and *Sulfuricurvum*) constituted 9.0%, while their abundance in other samples did not exceed 0.65%. The share of Deltaproteobacteria (genera *Desulfuromonas* and *Geopsychrobacter*) was 11.6%. *Desulfuromonas* was not detected in the water column of the studied lakes, and its abundance in the bottom sediments did not exceed 0.81%.

**4    Discussion**

This paper aimed to quantify the microbial processes within the water column and lake sediments of two different types of lakes widely presented in Yamal peninsula: deep mature lakes with established basins and small shallow lakes. In the first case, the basins likely originated as a result of thermokarst on massive ground ice as the basin was embedded into all geomorphological levels. In the second case, thermokarst on constitutional ground ice might be a starting mechanism for the
formation of lake basins. However, other mechanisms responsible for the formation of these lakes are plausible.

Microorganisms of the community developing in the water column of tundra lakes during the summer season may be considered as members of four groups: (1) allochthonous permafrost microbiome delivered to the lake with thawed ground ice (2) autochthonous photosynthetic microorganisms (cyanobacteria and algae detected as algal chloroplasts), with a characteristic brief bloom; (3) heterotrophic (chemoorganotrophic) bacteria consuming dissolved and particulate organic
matter; and (4) methano- and methylotrophs, as well as chemolithotrophic bacteria consuming methane and other reduced compounds released from the bottom sediments.

**4.1    Taxonomic composition of microbial communities in the water column**

The community of the first (allochthonous) group was genetically diverse. It contained heterotrophs of several taxonomic groups which were almost absent in the samples of suspended matter and bottom sediments from all the studied lakes.
The second group (photosynthetic microorganisms) contained cyanobacteria related to the families Nostocacea and Cyanobiaceae. In the tundra of northern Québec, Canada, phototrophic microorganisms were represented by picocyanobacterial groups Synechococcales and Nostocaceae, the nitrogen-fixing species *Dolichospermum curvum* (Wacklin et al. 2009), formerly known as *Anabaena curva* (Crevecoeur et al., 2015). At deeper parts of Canadian lakes, an abundance of the OTUs of anoxygenic green bacteria of the genus *Pelodictyon* (Chlorobiaceae) was as high as 50%. Our study revealed
no Chlorobiaceae in either the shallow thermokarst or deep lakes.





Among heterotrophic bacteria (the third group), the dominant phyla were Actinobacteria (18.4–30.7%) and Bacteroidetes (6.2–24.4%), including such species as actinobacteria "*Candidatus* Planktophila" and "*Candidatus* Nanopelagicus," as well as *Flavobacterium succinicans*, and *Sediminibacterium* sp. of the phylum *Bacteroidetes.*

Since tundra lakes are sources of atmospheric methane (Laurion et al., 2010), the composition of the water column methanotrophic community is of special interest. Because of their ability to oxidize methane, aerobic methanotrophs can significantly reduce methane emissions to the atmosphere and thus play an important role in the global methane cycle (Conrad, 1996). Aerobic methanotrophic bacteria belong to three main lineages: Betaproteobacteria, Alphaproteobacteria, and Verrucomicrobia, with different carbon assimilation pathways (Knief, 2015). In the water column of all studied lakes, methanotrophic microorganisms were far from numerous (< 0.2%) and belonged exclusively to the order Methylococcales

(class Gammaproteobacteria). Most of them belonged to the genus *Methylobacter*. The bacteria detected in the water samples included members of the genera *Methylophilus* (methanol-oxidizing bacteria), *Acinetobacter* (known to perform denitrification coupled to oxidation of various organic compounds, including humic and fulvic acids) (Wen et al., 2019), and *Limnohabitans*, as well as "*Candidatus* Methylopumilus turicensis." The latter are methylotrophic planktonic psychrophilic bacteria capable of methanol and methylamine oxidation and preferring the low-temperature conditions of the hypolimnion (Salcher et al.,

345     2015).

## 4.2  Taxonomic composition of microbial communities in the bottom sediments

Gammaproteobacteria of the family Methylomonaceae were found in the sediments of all four lakes (up to 5.2%). The sequences detected were most closely related to those of *Methylobacter tundripaludum* of the family Methylomonaceae (Smith and Wrighton, 2019). They were most abundant in the upper sediment layer and could account for the observed

methane oxidation. Members of this family can grow under low oxygen conditions and act as major methane consumers in stratified lakes (Oswald et al., 2017).

Analysis of the published data indicates that members of the families Methylomonaceae and Methylococcaceae predominate in many thermokarst basins (thaw ponds and lakes) in Alaska (de Jong et al., 2018; Matheus Carnevali et al., 2018) and Canada (Crevecoeur et al., 2015, 2017). *Methylobacter tundripaludum* is a typical component of microbial

communities in Arctic bogs and swamped tundra soils (Graef et al., 2011; Liebner et al., 2009; Wartiainen et al., 2006).

Comparison of the data on MO rates and *Methylobacter* relative abundance indicates that these bacteria probably lay the basis for the microbial community utilizing methane in the sediments of tundra lakes.

Methane oxidation can be done, apart from aerobic methanotrophic bacteria, by the ANME-2d group archaea assigned to "*Candidatus* Methanoperedens." These archaea may perform anaerobic methane oxidation coupled to the reduction of nitrate

(Haroon et al., 2013), or iron and manganese oxides (Ettwig et al., 2016; Leu et al., 2020).

These organisms were detected in the sediments of all lakes, except for lake LK-003, and in some cases their share was as high as 3%. Thus, during the short warm period of the summer season, in all four tundra lakes, the microbial community of



the oxidative branch of the methane cycle contained both aerobic methanotrophic and methylotrophic bacteria and ANME-2d archaea.

365       In the absence of sulfate ions, methanogenesis is well known to be the terminal stage of OM degradation in the sediments. Methane formation in tundra lakes (either thermokarst ones or of other types) is known to occur via either the hydrogenotrophic or the aceticlastic pathway; their combination is also possible (de Jong et al., 2018; Matheus Carnevali et al., 2015, 2018). Members of the families Methanosarcinaceae and Methanosaetaceae have been most commonly detected in archaeal methanogenic communities of the bottom sediments of thermokarst ponds. Hydrogenotrophic members of the orders
Methanomicrobiales, Methanobacteriales, and Methanocellales were also detected (Matheus Carnevali et al., 2018; Negandhi et al., 2014). The predominance of members of certain taxonomic and physiological groups among the microorganisms of the methane cycle in different horizons of stratified basins has been reported for Lake Pavin (Biderre-Petit et al., 2019). In this meromictic lake, archaea of the genus *Methanoregula* predominate in the methanogenic community under anoxic conditions, while methanotrophic bacteria of the genus *Methylobacter* prevail both in the methanotrophic community of the oxic water
column and at the oxygen–methane interface.

       The methanogenic archaeal community in the sediments of the studied Yamal lakes was represented by several OTUs of *Methanoregula* (order Methanomicrobiales), which constituted up to 16% of all microorganisms. The abundance of the OTUs of *Methanosaeta* (order Methanosarcinales) in the sediments of LK-003 was up to 16.9%. The abundance of the OTUs of methylotrophic Euryarchaeota of the order Methanomassiliicoccales (Dridi et al., 2012) was up to 26% in LK-004 and up to
32% in LK-002 sediments.

       Yamal sediments differed significantly in the dominance and the ratio of the major archaeal taxa from those in the sediments of the Emaiksoun thermokarst lake, Utqiagvik, Alaska, where, apart from the OTUs of Methanosaetaceae (30–31%), high levels of Bathyarchaeota OTUs were revealed (17%–24%) (de Jong et al., 2018).

       Since only the rate of hydrogenotrophic methanogenesis (MG-h) was determined in the present work, the overall MG
rate is most probably an underestimate. This is indirectly indicated by our data on the high abundance of Methanosarcinales and Methanomicrobiales OTUs in the sediments of all four lakes. Members of these lineages are known to be equally capable of both aceticlastic and hydrogenotrophic methanogenesis (Crevecoeur et al., 2016). The contribution of aceticlastic methanogenesis may be rather significant. Thus, in Lake Rotsee (Switzerland) over 90% of methane released from the sediment into the water column was produced by aceticlastic *Methanosaeta* spp., with hydrogenotrophic methanogens responsible only
for 7% (Zepp Falz et al., 1999).

       Thus, our calculations estimate the amount of methane produced by archaea via the hydrogenotrophic pathway on the 0-15-cm sediment layer to be in the range 0.6 to 3.6 $\mu$mol $CH_4$ $m^{-2}$ $day^{-1}$ in shallow lakes and 3.5 to 14.5 $\mu$mol $CH_4$ $m^{-2}$ $day^{-1}$ in deep mature lakes. The estimated amount of methane oxidized in these sediments is 310–650 $\mu$mol $CH_4$ $m^{-2}$ $day^{-1}$ for shallow lakes and 300–1350 $\mu$mol $CH_4$ $m^{-2}$ $day^{-1}$ for deep mature lakes. Methane is released from the sediments into the water
column and is oxidized in the course of its diffusion from the bottom to the lake surface. The estimated amount of methane





oxidized in the water column is 7.2–15.5 µmol CH$_4$ m$^{-2}$ day$^{-1}$ for shallow lakes and 120–330 µmol CH$_4$ m$^{-2}$ day$^{-1}$ for deep mature.

### 4.3    Carbon Isotopic Composition of Methane, and Methane Oxidation Rate

The isotope composition of methane carbon (δ$^{13}$C-CH$_4$) in the bottom sediments (Table 3) correlated well with the MO rates
(Fig. 8).

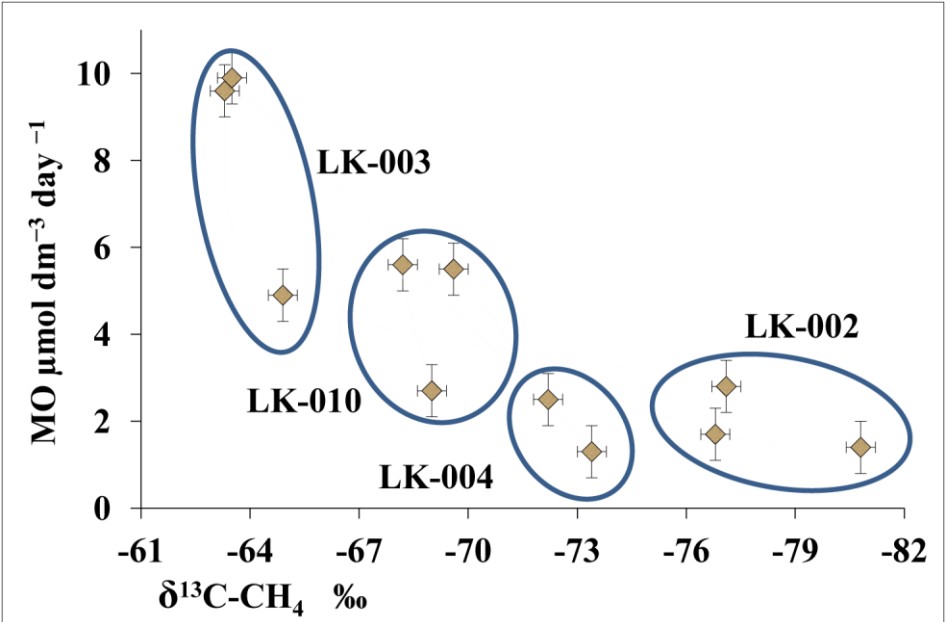

**Fig. 8. Isotopic composition of methane carbon (δ$^{13}$C-CH$_4$, ‰) in the bottom sediments of four Yamal lakes and the rates of methane oxidation (MO, µmol dm$^{-3}$ day $^{-1}$).**

High MO rates were associated with elevated levels of the heavy carbon isotope (LK-003, 3 samples), while low MO
rates were found in the sediments with a high content of the light carbon isotope (LK-002, 3 samples) (Fig. 8). The high content of the light carbon isotope in methane from all bottom sediments indicates its biogenic origin. Isotope analysis is, however, unable to differentiate between the modern biogenic methane and methane potentially released from thawing permafrost. MO results in preferential consumption of the light carbon isotope, with the remaining methane enriched with the heavy isotope; this was especially evident on the LK-003 sediments. At low MO rates, most of the overall methane remains unused, and the
resultant methane retains high levels of light carbon (LK-002). The low MO rate in the presence of sufficient methane concentrations indicates that the process is limited by some factors other than the substrate deficiency. This is probably the result of a limited supply of oxygen or electron acceptors other than oxygen, or the limited supply of biogenic elements.





## 4.4 Comparison of the activity of microbial processes in the summer and winter seasons

Since our study was carried out in the polar summer, during the short season of the high rate of all microbial processes,
comparison with the data obtained in April 2018 (Savvichev et al., 2018a) is required to understand the scale of seasonal variations. The data on the rates of microbial processes during the summer and winter seasons are presented in Table 4.

**Table 4. Methane concentrations and rates of microbial processes of the methane cycle in the near-bottom water layer and bottom sediments of Yamal lakes in April 2018 (Savvichev et al., 2018a) and August 2019**

|  | Bottom sediments (upper layer) | | | | Near-bottom water | | |
|---|---|---|---|---|---|---|---|
|  | $CH_4$, $\mu mol$ $dm^{-3}$ | MG-h, $nmol\ dm^{-3}$ $day^{-1}$ | MO, $\mu mol\ dm^{-3}$ $day^{-1}$ | $\delta^{13}C\text{-}CH_4$ ‰ | $CH_4$, $\mu mol\ L^{-1}$ | MO, $nmol\ L^{-1}$ $day^{-1}$ | $\delta^{13}C\text{-}CH_4$, ‰ |
| April 2018 | 45−450 | 2−48 | 0.05−0.18 | -89.1− -80.1 | 3.1−9.1 | < 2 | -77.1− -72.1 |
| August 2019 | 33−590 | 5−35 | 2−9.9 | -80.8− -64.9 | 0.3−4.1 | 3.6−55.3 | -65.6 |

420         MO − methane oxidation;        MG-h − hydrogenotrophic methanogenesis

A comparison of these data shows insignificant differences between methane concentrations in the upper sediment layers of Yamal lakes in summer and winter. The differences in the rates of hydrogenotrophic methanogenesis (MG-h) were also insignificant. MO rates were, however, 20–50 times higher in summer. This elevated MO rate resulted in higher levels of $^{13}C$-
$CH_4$ ($\delta^{13}C\text{-}CH_4$ to 64.9 ‰) in the residual (unconsumed) methane. The methane carbon isotope composition in winter was lighter and more homogeneous. MO rate in the water column was very low in winter, which was caused by the temperature decrease to 0–4°C, rather than the substrate limitation. The combination of low MO rates in winter and the formation of a thick ice cover resulted in increased methane concentrations in the water column under the ice. The highest seasonal methane concentration in the water column occurs probably immediately before the thawing of the lake ice cover. During the short
summer season, phytoplankton production in the Yamal tundra lakes of various depth is relatively high (340−1200 mg C m$^{-2}$ day$^{-1}$). Allochthonous and autochthonous organic matter partially degrades in the water column and precipitates onto the bottom sediments, where methane is the terminal product of anaerobic degradation (90−1000 $\mu mol\ CH_4\ dm^{-3}$).

Our analysis reveals the following scheme of summer microbial processes within the water column and lake sediments of two significantly different lake types: deep mature lakes (with established stable basins) and shallow lakes (resulted from the active
thermokarst on constitutional ground ice) (Fig. 9). OM decomposition occurs throughout the year. However, the MO rates in winter are low, which may result in methane accumulation within the water column. During the summer season, the major part of methane in deep mature lakes is oxidized in the water column, and low emission rates form the surface of these lakes may



be expected. Most of the area of shallow lakes is frozen to the bottom during winter, and allochthonous OM remains mainly undecomposed. We therefore observed lower rates of methane production in shallow lakes during summer, but since the MO rates were also lower, we expect similar dissolved methane concentrations and methane emissions from the surface of these shallow and deep mature lakes. Given that most of the water within deep lakes remains unfrozen during winter, we might expect that while the annual methane in these lakes is higher, their water column serves as a microbial filter for methane emission into the atmosphere.

The bottom sediments of tundra lakes are sources of methane, which is of biogenic origin, according to the data on its carbon isotope composition. The rates of hydrogenotrophic methanogenesis are higher in the sediments of deeper lakes than in the sediments of shallow thermokarst ones. In the sediments of both deep and shallow Yamal lakes, *Methanoregula* and *Methanosaeta* were predominant components of archaeal methanogenic communities (Fig. 9).

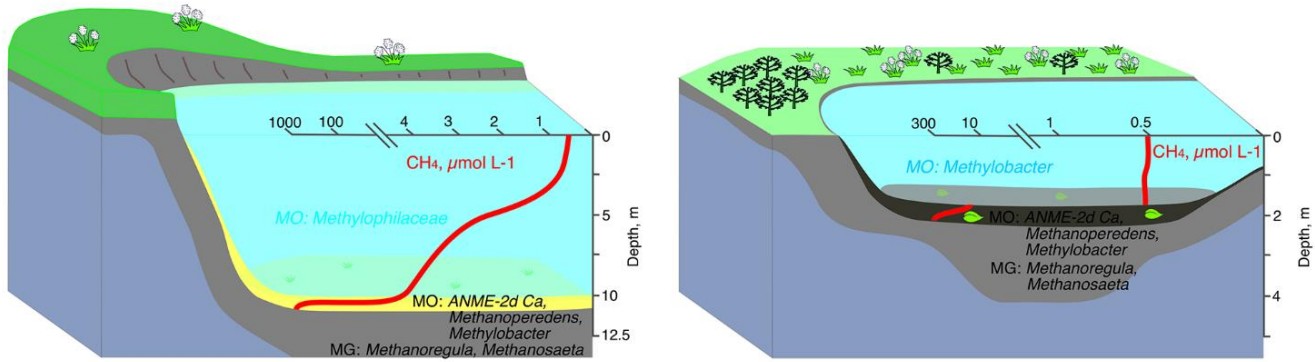

**Figure 9. Microbial processes and microbial communities of the methane cycle in the water columns and bottom sediments of deep (left) and shallow tundra lakes (right) of the Yamal Peninsula.**

In the sediments of lakes LK-002 and LK-004, the percentage of methylotrophic Methanomassiliicoccales was significant (OTUs up to 29% and 14%, respectively). In parallel to methanogenesis, methane oxidation occurs at rather high rates in the sediments of the studied lakes. MO is due to both ANME-2d archaea ("*Ca. Methanoperedens*") and bacteria of the genus *Methylobacter*, (OTUs up to 5.2% of the total number of reads) (Fig. 9). Methane not consumed in the sediments is released into the water column, where it is oxidized by the community of aerobic methano- and methylotrophic bacteria, in which methylotrophic betaproteobacteria of the family Methylophilaceae predominate. A decrease in the concentration of dissolved methane from 0.8−4.1 μmol $CH_4$ $L^{-1}$ in the near-bottom layer to 0.4 μmol $CH_4$ $L^{-1}$ at the surface is the visible geochemical result of the activity of methanotrophic bacteria in deep lakes(Fig. 9). In shallow lakes wind-mixed to the bottom, methanotrophic bacteria occur in trace amounts, their activity is low, and the geochemical methanotrophic effect is weak. Thus, the depth of a tundra lake is an important factor affecting the scale and completeness of the geochemical processes of the methane cycle caused by microbial activity. The role of the water column as a microbial filter is therefore significant.

## 5 Conclusions

The water column of tundra lakes is a transit zone for the movement of terrigenous and autochthonous organic matter into the bottom sediments. In the sediments, organic matter undergoes a deep microbial transformation, resulting in the formation of biogenic methane. Methane is released from the bottom sediments into the water column and flows to the water surface and further into the atmosphere. The microbial community in the water column consumes both organic matter and methane. Lake depth plays a significant role in the completeness of the substrate consumption. During the winter-spring season, sub-ice

conditions are characterized by accumulation of methane in the water column. In the short summer season, its consumption increases sharply. In deep water bodies, the methanotrophic microbial community is a natural filter that prevents methane release into the atmosphere. In shallow thermokarst lakes, the methanotrophic microbial filter is significantly less efficient due to the low thickness of the water layer, as well as wind mixing throughout the entire water column, which accelerates the transport of methane from the bottom to the surface.

## 6 Data availability


The raw data generated from 16S rRNA gene sequencing were deposited in Sequence Read Archive (SRA) under the accession numbers SRR11972844 -SRP266728, available via BioProject PRJNA636944.

## 7 Author contribution

Conceptualization: A.S. and M.L.; Methodology, A.S. and M.L.; Field Investigation: A.S., V.K., A.C., Y.D., and A.K. (Artem
Khomutov); Laboratory Investigation: A.S., I.R., E.V., and V.K.; Resources: N.P.; Writing—Original
Draft: A.S., M.L., and A.K. (Anna Kallistova); Writing—Review & Editing: A.S., M.L., A.K. (Anna Kallistova), and P.S.;
Supervision: N.P. and N.R.; Project Administration: N.P. and M.L.

## 8 Competing interests

The authors declare that they have no conflict of interest.

## 9 Financial Support

The authors thank the "Russian Center for Arctic Exploration" for organizing and supporting the fieldwork. Lake bathymetry, morphological analyses, and DOC measurements in the laboratory were funded by the Russian Foundation for Basic Research,
project 18-05-60222 and MK-3751.2019.5. Radiotracer studies were supported by the Russian Foundation for Basic Research,



project 20-04-00487. Investigation of the processes of the methane cycle and the composition of the microbial community was supported by the Russian Science Foundation, project 16-14-10201, as well as by the Ministry of Science and Higher Education of the Russian Federation.

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

**Figure Captions**

**Figure 1. A: An overview map of Eurasia, permafrost extent from (Brown et al., 2002). B: Study area map of Central Yamal, with red outlined polygons indicating the lakes under study, yellow star – Vas'kiny Dachi research station, the orthorectified QuickBird satellite image acquired July 30, 2010 as a background (source: Digital Globe Foundation ©), datum – WGS-84, projection – UTM Zone 42N. C: the topography of the lake basins under study, red dots indicate locations of the water and sediment samples, elevations are given in the Baltic height system (1977).**

**Figure 2. Primary production in four lakes of the Yamal Peninsula (August 2019): in the upper water horizon, µg C L$^{-1}$ day$^{-1}$ (1), and integral PP for the photic layer, mg C m$^{-2}$ day$^{-1}$ (2).**



**Figure 3. DOC concentration in the water column (a) and bottom sediments (b) of four lakes.**

**Figure 4.  Dark CO₂ assimilation (DCA) in four lakes of the Yamal Peninsula (August 2019): in the upper water horizon, µg C L⁻¹ day⁻¹ (1), and integral DCAint for the water column, mg C m⁻² day⁻¹ (2).**

**Figure 5. Concentrations of dissolved methane (CH₄, nmol L⁻¹) in the water columns of four Yamal tundra lakes: LK-002 (red), LK-010 (yellow), LK-003 (purple), and LK-004 (green).**

**Figure 6. Rates of methane oxidation, nmol CH₄ L⁻¹ day⁻¹, in the water column of four Yamal tundra lakes: LK-002 (red), LK-010 (yellow), LK-003 (purple), and LK-004 (green).**

**Figure 7.  Microbial communities of the water column and bottom sediments in four tundra Lakes of Yamal peninsula determined**
**by high-throughput sequencing of 16S rRNA genes.**

**Figure 8. Isotopic composition of methane carbon (δ¹³C-CH₄, ‰) in the bottom sediments of four Yamal lakes and the rates of methane oxidation (MO, µmol dm⁻³ day ⁻¹).**

**Figure 9.  Microbial processes and microbial communities of the methane cycle in the water columns and bottom sediments of deep (left) and shallow tundra lakes (right) of the Yamal Peninsula.**
