# Peer review of "The water column of the Yamal tundra lakes as a microbial filter preventing methane emission"

_Biogeosciences, 2020_

## Referee Comment (RC1) · June Skeeter (Referee) · 23 Nov 2020

**Review of: The water column of the Yamal tundra lakes as a microbial filter preventing methane emission**

**General Comments:**

This manuscript contains information that appears to be of considerable value in understanding the role of methane production and consumption in both deep and shallow Yamal tundra lakes. It likely has valuable application to understanding these processes in thermokarst lakes across the arctic. However, there are some significant adjustments/edits needed.

The manuscript needs a thorough proof read. There are many grammatic errors/issues, I've outlined a fair number in the technical corrections, but this is by no means comprehensive.

Further, this manuscript needs to be more focused and organized. A more clearly outlined hypothesis/research statement at the end of the introduction would be helpful. There is a wealth of information presented here, but it is not immediately clear how some information relates to the stated goals. E.g. the information presented in section 3.1 seems to be fairly important to the processes involved in methane production, but there is minimal discussion of these results in section 4.

**Specific Comments:**

**Line 42:** Thermokarst lakes are also widespread in Northwestern Canada and the Hudson Bay Lowlands – eg. Marsh et al. (2009): Marsh, P., Russell, M., Pohl, S., Haywood, H. and Onclin, C.: Changes in thaw lake drainage in the Western Canadian Arctic from 1950 to 2000, Hydrol. Process., 23(1), 145–158, doi:10.1002/hyp.7179, 2009.

**Table 1:** This is a small sample size of lakes, and characteristics appear to be quite different depending on the lake, especially temperature. I think some discussion of possible reasons for these differences is merited. Also, I have a hard time believing the bottom temperatures for the deep lakes? How is it that LK-004 has a temperature of 14.2 ˚C at a depth of 11m while LK-002 is only 11.7 ˚C at the surface? This seems like a very weak temperature gradient for 11m of depth? Do you have temperatures (and other variables) available for all sample depths? It would be helpful if they were all presented here.

**Fig 2 & Lines 206-208**: Define what is considered to be the photic layer. Is this the integral of the entire water column? In line 207, the term photic depth is used instead? Also, it is not apparent from Fig 2 that LK-003 has higher PP than LK-002 as claimed? This requires further explanation.

**Line 209 & Fig 3a:** LK-004 appears to contradict the claim of < (± 0.5 mg L−1) variability in DOC within the water column?

**Line 203 & Fig 4:** Should this figure show the DCA values for the near-bottom layer of the lakes as well for better comparison?

**Fig 2 & 4:** The integrated values would be better shown in a separate subplot or with a secondary y-axis to make it more apparent they are in different units.

**Line 315:** What are the other mechanisms of formation?

**Lines 384 – 397:** I think this point needs further clarification? Why did you only measure hydrogenotrophic methanogenesis if it is likely to be only a small fraction of total methanogenesis? The methane production vs. oxidation figures presented in here make it seem like there should be no methane emissions because rates of oxidation are orders of magnitude higher than production?

**Line 471-474:** From the results presented here is it possible to get some sense of the relative magnitude methane emissions from the surfaces deep vs. shallow lakes? What are the implications of these findings in regards to climate change?

**Technical Corrections:**

**Fig 1, Table 1, & throughout text:** Would it not be better to refer to LK-010 as LK-001?

**Line 22:** (90−1000 μmol CH4 dm−3) – What timeframe is this over? One day? The whole summer? Clarify the temporal unit.

**Line 37:** Consider rewording- e.g. These lakes have been classified as thermokarst lakes in continuous ice-rich permafrost (Dubikov, 1982) although other origins have also been proposed (Arctic and Antarctic Research Institute, 1977; Kritsuk, 2010).

**Line 41:** Change "in case if the topography of the area is flat" to "in flat areas".

**Line 47:** Change "the increase of total lake area by 12% is observed" to "an increase in total lake are of 12% has been observed"

**Line 70:** Change "are usually revealed" to "usually occur"

**Line 76:** Define acronym OM

**Line 78:** Remove the word therefore

**Line 80:** Change "widely presented" to widespread

**Table 1:** Define or be clearer with use of acronyms (e.g. NL/WL – Lat/Lon). EC, this acronym isn't used in the text, but you spell out electrical conductivity on line 134. Be consistent. What is secci depth? Also, LK-004 is missing a ")" after the depth in the left-most column.

**Line 260:** This sentence as is reads like it might be more appropriate in section 2.2. You could reword to say something like "Bottom sediment samples from the bottom surface and to the depth of 14−15 cm are described in Table 3."

**Line 273-274:** Be consistent with acronym usage, you use methane oxidation in one sentence then MO in the next.

**Line 442:** annual methane what? Production? Emission?

---

## Author Comment (AC1) · 1 Dec 2020

Review of: The water column of the Yamal tundra lakes as a microbial filter preventing methane emission General Comments: This manuscript contains information that appears to be of considerable value in understanding the role of methane production and consumption in both deep and shallow Yamal tundra lakes. It likely has valuable application to understanding these processes in thermokarst lakes across the arctic. However, there are some significant adjustments/edits needed. The manuscript needs a thorough proof read. There are many grammatic errors/issues, I've outlined a fair number in the technical corrections, but this is by no means comprehensive. Further, this manuscript needs to be more focused and organized. A more clearly outlined hypothesis/research statement at the end of the introduction would be helpful. There is

a wealth of information presented here, but it is not immediately clear how some information relates to the stated goals. E.g. the information presented in section 3.1 seems to be fairly important to the processes involved in methane production, but there is minimal discussion of these results in section 4.

We thank the reviewer for carefully reviewing our manuscript.

Specific Comments: Line 42: Thermokarst lakes are also widespread in Northwestern Canada and the Hudson Bay Lowlands – eg. Marsh et al. (2009): Marsh, P., Russell, M., Pohl, S., Haywood, H. and Onclin, C.: Changes in thaw lake drainage in the Western Canadian Arctic from 1950 to 2000, Hydrol. Process., 23(1), 145–158, doi:10.1002/hyp.7179, 2009.

Introduced into the text on the advice of the reviewer

Table 1: This is a small sample size of lakes, and characteristics appear to be quite different depending on the lake, especially temperature. I think some discussion of possible reasons for these differences is merited. Also, I have a hard time believing the bottom temperatures for the deep lakes? How is it that LK-004 has a temperature of 14.2 ÌŁC at a depth of 11m while LK-002 is only 11.7 ÌŁC at the surface? This seems like a very weak temperature gradient for 11m of depth? Do you have temperatures (and other variables) available for all sample depths? It would be helpful if they were all presented here.

Explanation concerning the absence of stratification in the water column of deep lakes was added to the text.

Fig 2 & Lines 206-208: Define what is considered to be the photic layer. Is this the integral of the entire water column? In line 207, the term photic depth is used instead? Also, it is not apparent from Fig 2 that LK-003 has higher PP than LK-002 as claimed? This requires further explanation.

Fig. 2 was changed according to the Reviewer's recommendations. PP values for all

tested water horizons are provided. It may be seen that in shallow lakes photosynthetic production was detected at all depths, including the near-bottom horizon. The terms photic layer and photic depth were removed as hindering the understanding of the experimental procedure. Integral PP values are shown separately.

Line 209 & Fig 3a: LK-004 appears to contradict the claim of $< (\pm 0.5 \text{ mg L}{-1})$ variability in DOC within the water column?

Đąorrections made to figure 3A

Line 203 & Fig 4: Should this figure show the DCA values for the near-bottom layer of the lakes as well for better comparison?

Fig. 4 was changed according to the Reviewer's recommendations. DCA values for all tested water horizons, including the near-bottom ones, are provided.

Fig 2 & 4: The integrated values would be better shown in a separate subplot or with a secondary y-axis to make it more apparent they are in different units.

On Figs. 2 and 4 integral values of PP and DOC are shown separately, and their different units are noted.

Line 315: What are the other mechanisms of formation?

A brief description of other mechanisms of formation has been added to the manuscript.

Lines 384 – 397: I think this point needs further clarification? Why did you only measure hydrogenotrophic methanogenesis if it is likely to be only a small fraction of total methanogenesis? The methane production vs. oxidation figures presented in here make it seem like there should be no methane emissions because rates of oxidation are orders of magnitude higher than production?

We agree that this comment of the reviewer is very significant. Indeed, on the basis of our radioisotope studies, it is impossible to carry out a full-fledged balance calculation

of methane production. Therefore, we restrict ourselves to the following conclusion. ÂńBased on the above calculations, it can be concluded that the contribution of hydrogenotrophic methane to the total methane production in the upper sediment layer does not exceed 5%Âż.

Line 471-474: From the results presented here is it possible to get some sense of the relative magnitude methane emissions from the surfaces deep vs. shallow lakes? What are the implications of these findings in regards to climate change?

Indeed, the main superconclusion of our studies is that the microbial community of the water column of deep lakes is a fairly effective gas filter. The efficiency of methane utilization in the water column of thermokarst lakes is lower. It can be assumed that climate warming will lead to an increase in the total area of thermokarst lakes, which will enhance the effect of methane release into the atmosphere. To carry out quantitative calculations, it is required to use other methods than we use, namely the use of floating cameras. We believe this is the subject of another study.

Technical Corrections: Fig 1, Table 1, & throughout text: Would it not be better to refer to LK-010 as LK-001?

The digital names of the lakes are taken from a large database used by Yamal cryogeologists. In this database, a lake named LK-001 already exists.

Line 22: ($90-1000$ $\mu$mol ĐąĐİ4 dm$-3$) – What timeframe is this over? One day? The whole summer? Clarify the temporal unit.

Changed. The data refer to the concentration of dissolved methane, not to its production.

Line 37: Consider rewording- e.g. These lakes have been classified as thermokarst lakes in continuous ice-rich permafrost (Dubikov, 1982) although other origins have also been proposed (Arctic and Antarctic Research Institute, 1977; Kritsuk, 2010).

Changed

Line 41: Change "in case if the topography of the area is flat" to "in flat areas".

Changed

Line 47: Change "the increase of total lake area by 12% is observed" to "an increase in total lake are of 12% has been observed"

Changed

Line 70: Change "are usually revealed" to "usually occur"

Changed

Line 76: Define acronym OM

Definition has been provided above (line 55). Used this acronym several times more.

Line 78: Remove the word therefore

Removed

Line 80: Change "widely presented" to widespread

Changed to "common"

Table 1: Define or be clearer with use of acronyms (e.g. NL/WL – Lat/Lon). EC, this acronym isn't used in the text, but you spell out electrical conductivity on line 134. Be consistent. What is secci depth? Also, LK-004 is missing a ")" after the depth in the left-most column.

Corrected

Line 260: This sentence as is reads like it might be more appropriate in section 2.2. You could reword to say something like "Bottom sediment samples from the bottom surface and to the depth of 14−15 cm are described in Table 3."

Changed

Line 273-274: Be consistent with acronym usage, you use methane oxidation in one sentence then MO in the next.

Corrected

Line 442: annual methane what? Production? Emission?

Corrected

The figures changed according to the Reviewer's recommendations (Figs. 2, 3A, 3B, and 4) are uploaded.

[Figure]

[Figure]

**Fig. 1.** Fig. 2. Primary production (PP) in four lakes of the Yamal Peninsula (August 2019) in the water horizons, $\mu$g C L−1 day−1; and integral PP for the entire water column, mg C m−2 day−1.

[Figure]

**Fig. 2.** Fig. 3. DOC concentration in the water column (a) and bottom sediments (b) of four lakes.

[Figure]

**Fig. 3.**

**Fig. 4.** Fig. 4. Dark CO2 assimilation (DCA) in four lakes of the Yamal Peninsula (August 2019) in the water horizons, $\mu$g C L$-1$ day$-1$; and integral PP for the entire water column, mg C m$-2$ day$-1$.

---

## Referee Comment (RC2) · Anonymous Referee #2 · 20 Dec 2020

**Review of 'The water column of the Yamal tundra lakes as a microbial filter preventing methane emission'**

Alexander Savvichev, Igor Rusanov, Yury Dvornikov, Vitaly Kadnikov, Anna Kallistova, Elena Veslopolova, Antonina Chetverova, Marina Leibman, Pavel Sigalevich, Nikolay Pimenov, Nikolai Ravin, and Artem Khomutov

**Overview**

This manuscript deals with the differences in microbial processes, with focus on methane production and oxidation, between shallow and deep lakes of the Yamal tundra. Sampling was performed to measure methane concentration and stable isotopic signature in different depths as well as other environmental variables, and to collect samples for the characterization of microbial communities inhabiting the water column and sediments. Water and sediments samples were also taken for the determination of hydrogenotrophic methane production and methane oxidation rates using 14C labelled substrates ($NaH^{14}CO_3$ and $^{14}CH_4$). Light and dark CO2 assimilation incubations were also measured. Based on the measured rates of methane production and oxidation, on the stable isotopic signature of in situ methane and on the microbial communities detected in the sediments and water column, this study proposes an overview of the $CH_4$ cycle and microbial players in the two types (dep and shallow) of studied lakes. They conclude, based on the differences in the rates of methane production and consumption, as well as on the profiles of methane concentration, that the water columns of deeper tundra lakes are better methane filters than of the shallower lakes. The study provides a thorough investigation of C processes and microbial communities in the water and sediments and valuable information on these systems, but I believe some information is unnecessary to support the conclusions (ex: Fig 2, Fig 3, Fig 4, Table 2), creating detours in the main message of the manuscript. I think these should be better incorporated in the Discussion and Conclusion sections or removed from the manuscript. For example, Figure 9, which summarizes the main findings of the paper, does not show primary production, DOC, nor dark and light CO2 assimilation results. On the other hand, other information that are important to understand the differences between the systems and would support the conclusion, such as temperature profiles indicating the physical structure of the lakes, are missing. The manuscript also needs a grammar and spelling check. Therefore, I believe the manuscript needs a major revision.

**Specific comments:**

Introduction

The introduction can be more fluid and provide only information essential to the problem. For example, I think there are excessive descriptions on the origin of the lakes (e.g. lines 40-43 can be summarized; lines 47 and 48 could be removed).

Lines 48-50 could be removed since they do not provide necessary information to introduce the study.

Lines 55-61 can be summarized in fewer sentences. I agree that it is good to explain well the reasoning, but I would skip some of the detailed explanations, assuming that the readers are familiar with methane and carbon cycling in lakes.

Lines 66-67: please reformulate this sentence because in the way it is structured, it seems that you mean that methanogens also oxidize methane.

Lines 73-79: I suggest you summarize these sentences into one or two. You can assume that the readers are familiar with the stable isotopic fractionation of methane during microbial methane oxidation.

I think the Introduction should be restructured. Some parts should be removed to avoid excessive information that is not essential for the study question (e.g. excessive details on the origin of lakes, isotopic fractionation of methane during methane oxidation, origin of C for methane production). I also think that you should give more emphasis on the differences between the shallow and deep lakes that could lead to expected differences in methane cycling.

Caption of Fig 1 needs reformulation. Please break down the description of B into two or three sentences or use semicolons to clarify.

Methods

Table 1:

- Please indicate what EC stands for in a footnote of the table. What is 'sm$^{-1}$' in EC? Should it be $\mu S\ m^{-1}$?
- Typo in 'Secci', should be Secchi.
- What does N and T mean in the Type of lake? Please clarify.
- Same for IV and V in the Basin embedded in. Please clarify.
- Please add maximum depth of lakes in the table.
- Please replace 'Sampling horizons' for 'Sampling depths' since these are water samples (right?).

Lines 111-114: is this information on the ions essential to this work? It seems to me that it is not.

Lines 122-124: refer the reader to Table 1 instead of listing the sampled depth in the text.

Line 129: what is '(C)'?

Lines 152-155: break the sentence in two. There are two colons (:)

Line 156: sampling depths for incubations were the same described in Table 1? Please specify.

Line 161: "Incubation of water and sediment samples to determine the rates of other processes was also carried out in situ.". Please specify which other processes were measured in incubations in situ.

Line 161: Under which conditions did you keep the experiments for determination of MG? Were the sediments anoxic when sampled? How did you keep samples anoxic during incubations?

Lines 166-167: How did you separate $CO_2$, biomass and soluble organic matter in the methane oxidation experiments? Please clarify.

Lines 170-177: Not sure what is the difference between the description that starts in line 171 and the other starting in line 176. Maybe you should start with lines 176-177 and then explain how the isotopic composition was calculated based on the light and heavy isotopes concentrations (lines 171-175).

Results

Line 200: This sentence is not clear and needs rephrasing. Why is it the main characteristic of water bodies? Something like "PP is the main process of C fixation in lakes" would make more sense to me.

Line 208: …higher PP in **the** two deep mature lakes (Fig. 2)

Line 208: Please repeat here which lakes are the deep mature ones to help the readers.

Table 2: what is the unit of biomass? ug of what per L? Carbon? What was the conversion factor to go from cell volume (um3) to biomass? Please add it in the methods section.

Line 225: please indicate what are 'aggregated cells' and how you measure them in the methods section.

Figures 2 and 4: it would be better to create a legend for the bars fills (instead of using the numbers 1 and 2). Also, a secondary y axis would be more appropriate than showing both series of data in the same axis.

Line 230: please repeat here which one is the thermokarst lake.

Table 3: what is s1, s2, s3? Please clarify or remove them if not relevant.

Line 269: where is this sample in Table 3?

Lines 286-287: it should be like that right? Since the primer used targeted the bacterial 16S rRNA gene.

Figure 7: please indicate in the caption what 'w' and 's' after that name of the sample mean in the x axis.

Line293-294: did you use a primer for the bacterial 16S rRNA gene and another for the archaeal 16S rRNA gene? Or does the primer used cover both groups? Please clarify this in the Methods section and provide a reference for the primer's coverage of both groups if it is the case. It was not clear that you were evaluating both Bacterial and Archaeal communities.

Line 293: if you use the word 'significantly', please provide some statistical verification of the difference.

Line 337: known aerobic methanotrophs are in the Gammaproteobacteria, Alphaproteobacteria and Verrucomicrobia. There are not methanotrophs in the Betaproteobacteria class to my knowledges. Please verify and fix.

line 356-357: could you provide a correlation coefficient of some sort of quantitative estimation of such comparison?

Line 358: What about the C. *Methylomrabilis oxyfera* within the candidate phyla NC10? Ettwig et al 2015 and others.

Line 361 should be in the previous paragraph.

Line 367: typo, should be 'acetoclastic'

Line 387, 389: typos, should be 'acetoclastic'

Lines 391-397 should be in the Results section in my opinion.

Lines 391-397: How do you explain higher methane production rates than methane oxidation rates? Some discussion on this would be interesting.

Fig 8: the x axis should be inverted (the y axis should cross the x axis at the lowest (more negative) signature values).

Lines 407-410: not needed to repeat the enrichment in 13C-CH4 during methane oxidation here. It was already explained in the Methods section.

Section 4.4: the comparison between summer and winter is very interesting, but in my opinion is out of the scope of the manuscript. There are many interesting results in the ms already – and in my opinion some can be further discussed –, while the seasonal differences add other information that is not in the aim of this study. In addition, Table 4 shows data for all lakes together (if I understood correctly), which makes the reader wonder the differences between lakes since the focus of the study is the differences between deep and shallow lakes.

Figure 9: I cannot read the microbial player responsible for methane oxidation in the deep lakes. This figure is a nice summary of the results, maybe other components, if kept, should be included.

---

## Author Comment (AC2) · 11 Jan 2021

Dear reviewer, we are grateful for the great work you have done. We tried to take into account all comments and answer all questions. This manuscript deals with the differences in microbial processes, with focus on methane production and oxidation, between shallow and deep lakes of the Yamal tundra. Sampling was performed to measure methane concentration and stable isotopic signature in different depths as well as other environmental variables, and to collect samples for the characterization of microbial communities inhabiting the water column and sediments. Water and sediments samples were also taken for the determination of hydrogenotrophic methane production and methane oxidation rates using 14C labelled substrates (NaH14CO3 and 14CH4). Light and dark CO2 assimilation incubations were also measured. Based

on the measured rates of methane production and oxidation, on the stable isotopic signature of in situ methane and on the microbial communities detected in the sediments and water column, this study proposes an overview of the CH4 cycle and microbial players in the two types (dep and shallow) of studied lakes. They conclude, based on the differences in the rates of methane production and consumption, as well as on the profiles of methane concentration, that the water columns of deeper tundra lakes are better methane filters than of the shallower lakes. The study provides a thorough investigation of C processes and microbial communities in the water and sediments and valuable information on these systems, but I believe some information is unnecessary to support the conclusions (ex: Fig 2, Fig 3, Fig 4, Table 2), creating detours in the main message of the manuscript. I think these should be better incorporated in the Discussion and Conclusion sections or removed from the manuscript. For example, Figure 9, which summarizes the main findings of the paper, does not show primary production, DOC, nor dark and light CO2 assimilation results. On the other hand, other information that are important to understand the differences between the systems and would support the conclusion, such as temperature profiles indicating the physical structure of the lakes, are missing. The manuscript also needs a grammar and spelling check. Therefore, I believe the manuscript needs a major revision.

Our manuscript presents the data on the composition and activity of microbial communities, somewhat focused on the methane cycle. The research subjects were four lakes of the continuous permafrost zone. Prior to our studies, we had no idea of the scale of the effect of lake depth on the rates of the methane cycle processes. The composition of microbial communities of the methane cycle and the rates of the relevant microbial processes certainly depend on such parameters as primary production and qualitative and quantitative composition of organic matter. Depth was only one easily discernible factor affecting the rate of microbial methane oxidation. We expected organic matter produced during the summer algal bloom (Fig. 2) to be the first component of the carbon cycle. The concentration of dissolved organic matter in the water and sediments (Fig. 3) is certainly a factor affecting the rates of microbial processes of the methane

cycle. Total microbial abundance (Table 2) and the rate of dark $CO_2$ assimilation (Fig. 4) characterize the state of the microbial community. In our opinion, these parameters are required for assessment of the trophic status of the studied basins, and removal of this information from the manuscript will result in a loss of valuable data. We thank the Reviewer for his comment concerning the temperature profiles. These data will be added to the revised version of the manuscript.

Introduction The introduction can be more fluid and provide only information essential to the problem. For example, I think there are excessive descriptions on the origin of the lakes (e.g. lines 40-43 can be summarized; lines 47 and 48 could be removed).

In our opinion, the Introduction is written in the classical style. The text includes substantiation of the studied problem and a short description of the origin of tundra lakes. The publications reporting wide occurrence of thermokarst lakes are listed. The data on association between climatic changes and formation of new thermokarst lakes are cited. The Introduction section is 60 lines long, which in less than in most Biogeoscience publications. In our opinion, removal of lines 40-43, 47-50, 66-61, and 73-79 will not improve the Introduction.

66-67 Reformulated based on the reviewer's comment. Caption of Fig 1 Reformulated based on the reviewer's comment.

Methods Table 1: • Please indicate what EC stands for in a footnote of the table. What is 'sm-1 ' in EC? Should it be $\mu$S m-1 ? Corrected: Electrical Conductivity (EC), $\mu$S cm-1

• Typo in 'Secci', should be Secchi. Corrected.

What does N and T mean in the Type of lake? Please clarify. Removed.

• Same for IV and V in the Basin embedded in. Please clarify. Removed.

• Please add maximum depth of lakes in the table. Added in the first column of the table 1.

• Please replace 'Sampling horizons' for 'Sampling depths' since these are water samples (right?). Corrected.

Lines 111-114: is this information on the ions essential to this work? It seems to me that it is not.

In our opinion, the information on the ion composition is important. Ion ratio in fresh waters is known to vary, with higher $SO_4^{2-}$ concentration resulting in shifts in the microbial community composition and intensification of sulfate reduction.

Lines 122-124: refer the reader to Table 1 instead of listing the sampled depth in the text. Corrected.

Line 129: what is '(C)'? Removed.

Lines 152-155: break the sentence in two. There are two colons (:) Corrected.

Line 156: sampling depths for incubations were the same described in Table 1? Please specify. Corrected.

Line 161: "Incubation of water and sediment samples to determine the rates of other processes was also carried out in situ." Please specify which other processes were measured in incubations in situ. Added based on the reviewer's comment.

Line 161: Under which conditions did you keep the experiments for determination of MG? Were the sediments anoxic when sampled? How did you keep samples anoxic during incubations?

Sediment samples were collected from intact cores into cut-off syringes sealed with rubber stoppers, avoiding air inflow. The labeled compounds and the fixing agent (KOH) were injected through the rubber stopper. Thus, no contact between the samples and air occurred at any stage of the experiment. The procedure has been described in detail in Pimenov, N.V. and Bonch-Osmolovskaya, E.A., In situ activity studies in thermal environments, Methods in Microbiology, vol. 35, Extremophiles, Rainey,

F.A. and Oren, A., Eds., Amsterdam: Acad. Press, Elsevier, 2006, pp. 29–53.

Lines 166-167: How did you separate CO2, biomass and soluble organic matter in the methane oxidation experiments? Please clarify.

Detailed description of the procedure would have taken too much space. It has been described previously (Pimenov, N.V. and Bonch-Osmolovskaya, E.A., In situ activity studies in thermal environments, Methods in Microbiology, vol. 35, Extremophiles, Rainey, F.A. and Oren, A., Eds., Amsterdam: Acad. Press, Elsevier, 2006, pp. 29–53).

Lines 170-177: Not sure what is the difference between the description that starts in line 171 and the other starting in line 176. Maybe you should start with lines 176-177 and then explain how the isotopic composition was calculated based on the light and heavy isotopes concentrations (lines 171-175). Reformulated based on the reviewer's comment.

Results Line 200: This sentence is not clear and needs rephrasing. Why is it the main characteristic of water bodies? Something like "PP is the main process of C fixation in lakes" would make more sense to me. The Results section indeed begins with our results on the phytoplankton primary production. In our opinion, freshly produced organic matter of phytoplankton origin is the initial substrate for methanogenesis in the sediments of the studied lakes. We have rephrased the sentence according to the Reviewer's comments.

Line 208: . . .higher PP in the two deep mature lakes (Fig. 2) Corrected to "a little higher PP in the two deep mature lakes."

Line 208: Please repeat here which lakes are the deep mature ones to help the readers. Added.

Table 2: what is the unit of biomass? ug of what per L? Carbon? What was the conversion factor to go from cell volume (um3) to biomass? Please add it in the methods section. Biomass was calculated using the data on the volume of microbial cells

and assuming the density of wet biomass equal to 1.0 mg mm-3. Specific biomass of microbial cells (B) is therefore presented in $\mu$g L-1. Although carbon content in the biomass can be easily calculated using the known coefficient of 20 fg C per cell (Lee and Fuhrman, 1987), we do not think this is necessary, since the data will not be discussed further.

Line 225: please indicate what are 'aggregated cells' and how you measure them in the methods section. During microscopic examination of the stained preparations, single cells and cells associated with aggregates were enumerated separately. A group of cells with a common outline, in which enumeration of individual cells was difficult, was considered an aggregate.

Figures 2 and 4: it would be better to create a legend for the bars fills (instead of using the numbers 1 and 2). Also, a secondary y axis would be more appropriate than showing both series of data in the same axis. Figs. 2 and 4 were completely remade in accordance with the Reviewer's comments.

Line 230: please repeat here which one is the thermokarst lake. Added.

Table 3: what is s1, s2, s3? Please clarify or remove them if not relevant. S1, s2, and s3 are designations of the sediment samples, which are used also on Fig. 7. Abbreviations are explained in the caption to Table 3.

Line 269: where is this sample in Table 3? The error is corrected. Sample LK-004 K is shown in Table 3.

Lines 286-287: it should be like that right? Since the primer used targeted the bacterial 16S rRNA gene. Yes, the primers used in this work were amplified with both bacterial and archaeal groups.

Figure 7: please indicate in the caption what 'w' and 's' after that name of the sample mean in the x axis. W and S are designations for the water and sediment samples, respectively. The figure caption was modified accordingly.

Line293-294: did you use a primer for the bacterial 16S rRNA gene and another for the archaeal 16S rRNA gene? Or does the primer used cover both groups? Please clarify this in the Methods section and provide a reference for the primer's coverage of both groups if it is the case. It was not clear that you were evaluating both Bacterial and Archaeal communities. Yes, the primers used in this work were amplified with both bacterial and archaeal groups. The relevant reference was added: Frey B, Rime T, Phillips M, Stierli B, Hajdas I, Widmer F, and Hartmann M. Microbial diversity in European alpine permafrost and active layers. FEMS Microbiol Ecol. 2016; 92(3):fiw018. https://doi.org/10.1093/femsec/fiw018

Line 293: if you use the word 'significantly', please provide some statistical verification of the difference. Corrected.

Line 337: known aerobic methanotrophs are in the Gammaproteobacteria, Alphaproteobacteria and Verrucomicrobia. There are not methanotrophs in the Betaproteobacteria class to my knowledges. Please verify and fix. Corrected. It was our mistake, since there are no methanotrophs in the Betaproteobacteria class.

line 356-357: could you provide a correlation coefficient of some sort of quantitative estimation of such comparison? Comparison of the data on activity of methane oxidation and Methylobacter relative abundance is an estimate. We are not sure the correlation coefficients will affect our cautious conclusions.

Line 358: What about the C. Methylomrabilis oxyfera within the candidate phyla NC10? Ettwig et al 2015 and others. No other methanotrophic bacteria and archaea were detected, including the candidate phylum NC10.

Line 361 should be in the previous paragraph. The paragraphs were merged.

Line 367: typo, should be 'acetoclastic' Corrected. Both variants occur in the literature, although "acetoclastic" is indeed more common.

Line 387, 389: typos, should be 'acetoclastic' Corrected.

Lines 391-397 should be in the Results section in my opinion. In lines 391–397, the calculated integral data on methane production and oxidation in the upper sediment layer ($\mu$mol CH4 m$-2$ day$-1$) are presented. They were calculated for the 0-15 cm sediment layer. Since changing this range in any direction results in different calculated values, we think that the results of our calculations are not experimental results, but rather belong to the Discussion section.

Lines 391-397: How do you explain higher methane production rates than methane oxidation rates? Some discussion on this would be interesting. In the studied sediments, the rate of methane production (0.6–14.5 $\mu$mol CH4 m$-2$ day$-1$) was much lower than the rate of methane oxidation (300–1350 $\mu$mol CH4 m$-2$ day$-1$). This discrepancy may be caused either by low rates of hydrogenotrophic methanogenesis or by methane inflow from deeper sediment layers (below the studied horizons).

Fig 8: the x axis should be inverted (the y axis should cross the x axis at the lowest (more negative) signature values). The figure was modified according to the Reviewer's request.

Lines 407-410: not needed to repeat the enrichment in 13C-CH4 during methane oxidation here. It was already explained in the Methods section. Interpretation of the data on carbon isotope fractionation during microbial methane oxidation is not unequivocal. Fractionation depends on a number of parameters and can not be predicted from initial conditions. The text of lines 407–410 belongs to the Discussion section, since it considers the possible reasons for the changes in the carbon isotopic composition due to MO.

Section 4.4: the comparison between summer and winter is very interesting, but in my opinion is out of the scope of the manuscript. There are many interesting results in the ms already – and in my opinion some can be further discussed –, while the seasonal differences add other information that is not in the aim of this study. In addition, Table 4 shows data for all lakes together (if I understood correctly), which makes the reader

wonder the differences between lakes since the focus of the study is the differences between deep and shallow lakes. Research on the rates of microbial processes and the composition of microbial communities in the Yamal polar lakes in summer and winter was carried out by the same team using the same methods. This is probably the first seasonal study of such difficult-to access polar basins. Although comparison of the winter and summer data did not yet yield clear conclusions, it still retains importance.

Figure 9: I cannot read the microbial player responsible for methane oxidation in the deep lakes. This figure is a nice summary of the results, maybe other components, if kept,should be included. Fig. 9 was intended as a conclusion of the Discussion section. In our opinion, introduction of additional material will hinder its understanding.

[Figure]

**Fig. 1.** Fig. 2. Primary production in four lakes of the Yamal Peninsula (August 2019): in the water horizons, $\mu$g C L$-1$ day$-1$, and calculated integral values PP, mg C m$-2$ day$-1$ (in the center of the picture fi

[Figure]

**Fig. 2.** Fig. 4. Dark CO2 assimilation (DCA) in four lakes of the Yamal Peninsula (August 2019): in the water horizons, $\mu$g C L−1 day−1, and calculated integral values DCA, mg C m−2 day−1 (in the center of the

[Figure]

**Fig. 3.** Fig. 8. Isotopic composition of methane carbon ($\delta$ 13 C-CH 4 , ‰ in the bottom sediments of four Yamal lakes and the rates of methane oxidation (MO, $\mu$mol dm $-3$ day $-1$ ).

---

## Referee Report (RR1)

**Review of 'The water column of the Yamal tundra lakes as a microbial filter preventing methane emission'**

**By Alexander Savvichev, Igor Rusanov, Yury Dvornikov, Vitaly Kadnikov, Anna Kallistova, Elena Veslopolova, Antonina Chetverova, Marina Leibman, Pavel Sigalevich, Nikolay Pimenov, Nikolai Ravin, and Artem Khomutov**

I totally agree with the authors that information on primary production, concentration of dissolved organic matter, microbial abundance and CO2 assimilation is important to the characterization of the studied systems, particularly if these lakes have never been studied before, as pointed out by the authors. However, in my opinion, this information could be presented as supplementary information or even in a different manuscript. There is no doubt that these data bring valuable information, but I believe they are not crucial to the main conclusions of the ms. As an example, I pointed out that Fig. 9, which summarizes the main findings of the ms, does not show these data, suggesting that they are not critical to arrive to the conclusions. But I understand this may be an issue of style, as it is my preference to read and write more succinct papers that present only the necessary data that take the readers to the main messages that are being conveyed.

On a similar note, I also prefer introductions that are straighter to the point, but I understand that the authors may have a different writing style.

Please add the reference Pimenov et al. 2006 where applicable in the Methods so that the reader can refer to it for more detailed methodology.

Please add in the Methods section that single cells and aggregated cells were visually differentiated under the microscope.

Fig 2 and 4: what are the light brown horizontal bars? I believe the bar plots (as original) with legends indicating which bar corresponds to the values in $\mu gC\ L^{-1}\ d^{-1}$ and which bar corresponds to $mgC\ m^{-2}\ d^{-1}$ are more appropriated. There is no need for a secondary y axis if both have the same range and are identical. A secondary y axis could be useful here if you wanted to show the per litter PP in a narrower range axis than the axis of the depth weighted PP. See below two fictitious figures showing how I think you could present your figures to make them clearer (first with one y axis, second one with two y axes):

---

## Referee Report (RR2)

**Review of: The water column of the Yamal tundra lakes as a microbial filter preventing methane emission**

**General Comments:**

This manuscript contains information that appears to be of considerable value in understanding the role of methane production and consumption in both deep and shallow Yamal tundra lakes. It likely has valuable application to understanding these processes in thermokarst lakes across the arctic. However, there are some significant adjustments/edits needed. With some minor edits/changes, I feel it can be accepted.

**Specific Comments:**

Lines 93-95: As stated in my previous review, I think a more clearly outlined hypothesis/research statement at the end of the introduction would be helpful. There is a wealth of information presented here, but it is not immediately clear how some information relates to the stated goals. I would suggest using a number list, eg: "The present work was aimed at elucidation of the similarities and differences in the rates of the methane cycle processes by examining: 1) the rate of hydrogenotrophic methanogenesis, 2) the rate of methane oxidation, 3) …. Etc."

Line 165: In your response to my previous review, you stated "on the basis of our radioisotope studies, it is impossible to carry out a full-fledged balance calculation of methane production". I think it would be valuable to explicitly state this early on in the text. Be clear that only a small portion of the total methane production is actually being quantified here because any readers might not be aware of this.

Line 281: Table 3 – This is just showing hydrogenotrophic methanogenesis right? I assume yes? Clarify please. You use the symbol MG-h later in the text.

**Technical Corrections:**

Lines 44-47: "Thermokarst lakes are widespread in West and East Siberia, in Alaska and Northern Scandinavia (Grosse et al., 2013; Kravtsova and Rodionova, 45 2016; Vonk et al., 2015; Wik et al., 2016). Thermokarst lakes are also common in Northwestern Canada and the Hudson Bay Lowlands (Marsh et al., 2009)." Would read better as "Thermokarst lakes are widespread in West and East Siberia, in Alaska and Northwestern Canada, the Hudson Bay Lowlands, and Northern Scandinavia (Marsh et al., 2009; Grosse et al., 2013; Kravtsova and Rodionova, 45 2016; Vonk et al., 2015; Wik et al., 2016).

Lines 71-72: Consider changing: "they oxidize this greenhouse gas and decrease methane emission into the atmosphere" to "they oxidize methane, thereby decreasing emissions into the atmosphere"

Line 165: (MG) should be (MG-h) to be consistent with further down in the text (eg. line 407)

Lines 349 – 428: Paragraphs are indented, inconsistent w/ rest of the paper

---

## Author Response (AR2)

Reviewer 1
Review of 'The water column of the Yamal tundra lakes as a microbial filter preventing methane emission'
By Alexander Savvichev, Igor Rusanov, Yury Dvornikov, Vitaly Kadnikov, Anna Kallistova, Elena Veslopolova, Antonina Chetverova, Marina Leibman, Pavel Sigalevich, Nikolay Pimenov, Nikolai Ravin, and Artem Khomutov

I totally agree with the authors that information on primary production, concentration of dissolved organic matter, microbial abundance and CO2 assimilation is important to the characterization of the studied systems, particularly if these lakes have never been studied before, as pointed out by the authors. However, in my opinion, this information could be presented as supplementary information or even in a different manuscript. There is no doubt that these data bring valuable information, but I believe they are not crucial to the main conclusions of the ms. As an example, I pointed out that Fig. 9, which summarizes the main findings of the ms, does not show these data, suggesting that they are not critical to arrive to the conclusions. But I understand this may be an issue of style, as it is my preference to read and write more succinct papers that present only the necessary data that take the readers to the main messages that are being conveyed.
We agree that some of the parameters studied had little effect on methanogenesis and methane emission (as demonstrated by Fig. 9), but this was by no means evident at the onset of the study. In our opinion, it is desirable to keep this information in the text to show the readers how we arrived at our conclusions.

On a similar note, I also prefer introductions that are straighter to the point, but I understand that the authors may have a different writing style.
The work was intended as a microbiological and biogeochemical assessment of the Central Yamal thermokarst lakes. The long Introduction was required to describe the diverse aspects of microbial processes and communities in thermokarst lakes.

Please add the reference Pimenov et al. 2006 where applicable in the Methods so that the reader can refer to it for more detailed methodology.
Added (line 141).

Please add in the Methods section that single cells and aggregated cells were visually differentiated under the microscope.
Added (line 163).

Fig 2 and 4: what are the light brown horizontal bars? I believe the bar plots (as original) with legends indicating which bar corresponds to the values in µgC L -1 d -1 and which bar corresponds to mgC m -2 d -1 are more appropriated. There is no need for a secondary y axis if both have the same range and are identical. A secondary y axis could be useful here if you wanted to show the per litter PP in a narrower range axis than the axis of the depth weighted PP. See below two fictitious figures showing how I think you could present your figures to make them clearer (first with one y axis, second one with two y axes):
The brown horizontal bars were termed "wavy bands" in the caption, since they are indeed wavy and therefore not strictly horizontal. They indicate the bottom location. In our opinion, such presentation is preferable, since lake depth is one of the most important parameters. The word "brown" was added to the caption. Bar graphs were replaced with XY ones in order to show how the rates of microbial processes vary from horizon to horizon. The data on integral rates (mg C m$^{-2}$ d$^{-1}$) are presented on the figure as numeric values. The second Y axes were removed.

Reviewer 2

Review of: The water column of the Yamal tundra lakes as a microbial filter preventing
methane emission
General Comments:
This manuscript contains information that appears to be of considerable value in understanding the role of methane
production and consumption in both deep and shallow Yamal tundra lakes. It likely has valuable application to
understanding these processes in thermokarst lakes across the arctic. However, there are some significant adjust-
ments/edits needed. With some minor edits/changes, I feel it can be accepted.
Specific Comments:
Lines 93-95: As stated in my previous review, I think a more clearly outlined hypothesis/research statement at the
end of the introduction would be helpful. There is a wealth of information presented here, but it is not immediately
clear how some information relates to the stated goals. I would suggest using a number list, eg: "The present work
was aimed at elucidation of the similarities and differences in the rates of the methane cycle processes by examin-
ing: 1) the rate of hydrogenotrophic methanogenesis, 2) the rate of methane oxidation, 3) .... Etc."
The present work was aimed at microbiological and biogeochemical characterization of the carbon turnover in
Central Yamal small young lakes (constitutional ice thermokarst) and deep mature lakes (massive ground ice
thermokarst) with a focus on the methane cycle processes. For this purpose, investigation of the following
parameters was required: (1) primary production, (2) dark $CO_2$ assimilation, (3) the rate of hydrogenotrophic
methanogenesis, (4) the rate of methane oxidation, (5) abundance and composition of microbial communities in the
water column.

Line 165: In your response to my previous review, you stated "on the basis of our radioisotope studies, it is impossi-
ble to carry out a full-fledged balance calculation of methane production". I think it would be valuable to explicitly
state this early on in the text. Be clear that only a small portion of the total methane production is actually being
quantified here because any readers might not be aware of this.
Inserted (line 423).

Line 281: Table 3 – This is just showing hydrogenotrophic methanogenesis right? I assume yes? Clarify please. You
use the symbol MG-h later in the text.
Replaced globally.

Technical Corrections:
Lines 44-47: "Thermokarst lakes are widespread in West and East Siberia, in Alaska and Northern Scandinavia
(Grosse et al., 2013; Kravtsova and Rodionova, 45 2016; Vonk et al., 2015; Wik et al., 2016). Thermokarst lakes are
also common in Northwestern Canada and the Hudson Bay Lowlands (Marsh et al., 2009)." Would read better as
"Thermokarst lakes are widespread in West and East Siberia, in Alaska and Northwestern Canada, the Hudson Bay
Lowlands, and Northern Scandinavia (Marsh et al., 2009; Grosse et al., 2013; Kravtsova and Rodionova, 45 2016;
Vonk et al., 2015; Wik et al., 2016).
Corrected.

Lines 71-72: Consider changing: "they oxidize this greenhouse gas and decrease methane emission into the atmos-
phere" to "they oxidize methane, thereby decreasing emissions into the atmosphere"
Changed.

Line 165: (MG) should be (MG-h) to be consistent with further down in the text (eg. line 407)
Changed.
Lines 349 – 428: Paragraphs are indented, inconsistent w/ rest of the paper
Corrected